# Confident Natural Policy Gradient for Local Planning in $q_\pi$-realizable Constrained MDPs

**Tian Tian**
University of Alberta, Edmonton
`ttian@ualberta.ca`

**Lin F. Yang** *
University of California, Los Angeles
`linyang@ee.ucla.edu`

**Csaba Szepesvári** *
University of Alberta, Google DeepMind, Edmonton
`szepesva@ualberta.ca`

## Abstract

The constrained Markov decision process (CMDP) framework emerges as an important reinforcement learning approach for imposing safety or other critical objectives while maximizing cumulative reward. However, the current understanding of how to learn efficiently in a CMDP environment with a potentially infinite number of states remains under investigation, particularly when function approximation is applied to the value functions. In this paper, we address the learning problem given linear function approximation with $q_\pi$-realizability, where the value functions of all policies are linearly representable with a known feature map, a setting known to be more general and challenging than other linear settings. Utilizing a local-access model, we propose a novel primal-dual algorithm that, after $\tilde{O}(\text{poly}(d)\epsilon^{-3})$[1] queries, outputs with high probability a policy that strictly satisfies the constraints while nearly optimizing the value with respect to a reward function. Here, $d$ is the feature dimension and $\epsilon > 0$ is a given error. The algorithm relies on a carefully crafted off-policy evaluation procedure to evaluate the policy using historical data, which informs policy updates through policy gradients and conserves samples. To our knowledge, this is the first result achieving polynomial sample complexity for CMDP in the $q_\pi$-realizable setting.

## 1 Introduction

In the classical reinforcement learning (RL) framework, optimizing a single objective above all else can be challenging for safety-critical applications like autonomous driving, robotics, and Large Language Models (LLMs). For example, it may be difficult for an LLM agent to optimize a single reward that fulfills the objective of generating helpful responses while ensuring that the messages are harmless (Dai et al., 2024). In autonomous driving, designing a single reward often requires reliance on complex parameters and hard-coded knowledge, making the agent less efficient and adaptive (Kamran et al., 2022). Optimizing a single objective in motion planning involves combining heterogeneous quantities like path length and risks, which depend on conversion factors that are not necessarily straightforward to determine (Feyzabadi and Carpin, 2014).

The constrained Markov decision process (CMDP) framework (Altman, 2021) emerges as an important RL approach for imposing safety or other critical objectives while maximizing cumulative

---

* Corresponding author.

[1] Here $\tilde{O}(\cdot)$ hides $\log$ factors.

38th Conference on Neural Information Processing Systems (NeurIPS 2024).

reward (Wachi and Sui, 2020; Dai et al., 2024; Kamran et al., 2022; Wen et al., 2020; Girard and Reza Emami, 2015; Feyzabadi and Carpin, 2014).

In addition to the single reward function optimized under a standard Markov decision process (MDP), CMDP considers multiple reward functions, with one designated as the primary reward function. The goal of a CMDP is to find a policy that maximizes the primary reward function while satisfying constraints defined by the other reward functions. Although the results of this paper can be applied to multiple constraint functions, for simplicity of presentation, we consider the CMDP problem with only one constraint function.

Our current understanding of how to learn efficiently in a CMDP environment with a potentially infinite number of states remains limited, particularly when function approximation is applied to the value functions. Most works studying the sample efficiency of a learner have focused on the tabular or simple linear CMDP setting (see related works for more details). However, there has been little work in the more general settings such as the $q_\pi$-realizability, which assumes the value function of all policies can be approximated by a linear combination of a feature map with unknown parameters. Unlike Linear MDPs (Yang and Wang, 2019; Jin et al., 2020), where the transition model is assumed to be linearly representable by a feature map, $q_\pi$-realizability only imposes the assumption on the existence of a feature map to represent value functions of policies.

Nevertheless, the generality of $q_\pi$-realizability comes with a price, as it becomes considerably more challenging to design effective learning algorithms, even for the unconstrained settings. For the general online setting, we are only aware of one sample-efficient MDP learning algorithm (Weisz et al., 2023), which, however, is computationally inefficient. To tackle this issue, a line of research (Kearns et al., 2002; Yin et al., 2022; Hao et al., 2022; Weisz et al., 2022) applies the *local-access model*, where the RL algorithm can restart the environment from any visited states - a setting that is also practically motivated, especially when a simulator is provided. The local-access model is more general than the generative model (Kakade, 2003; Sidford et al., 2018; Yang and Wang, 2019; Lattimore et al., 2020; Vaswani et al., 2022), which allows visitation to arbitrary states in an MDP. The local-access model provides the ability to unlock both the sample and computational efficiency of learning with $q_\pi$-realizability for the unconstrained MDP settings. However, it remains unclear whether we can harness the power of local-access for CMDP learning.

In this paper, we present a systematic study of CMDP for large state spaces, given $q_\pi$-realizable function approximation in the local-access model. We summarize our contributions as follows:

- We design novel, computationally efficient primal-dual algorithms to learn CMDP near-optimal policies with the local-access model and $q_\pi$-realizable function classes. The algorithms can return policies with small constraint violations or even no constraint violations and can handle model misspecification.

- We provide theoretical guarantees for the algorithms, showing that they can compute an $\epsilon$-optimal policy with high probability, making no more than $\tilde{O}(\text{poly}(d)\epsilon^{-3})$ queries to the local-access model. The returned policies can strictly satisfy the constraint.

- Under the misspecification setting with a misspecification error $\omega$, we show that our algorithms achieve an $\tilde{O}(\omega) + \epsilon$ sub-optimality with high probability, maintaining the same sample efficiency of $\tilde{O}(\text{poly}(d)\epsilon^{-3})$.

## 2   Related works

Most provably efficient algorithms developed for CMDP are in the tabular and linear MDP settings. In the tabular setting, most notably are the works by (Efroni et al., 2020; Liu et al., 2021; Zheng and Ratliff, 2020; Vaswani et al., 2022; Kalagarla et al., 2021; Yu et al., 2021; Gattami et al., 2021; HasanzadeZonuzy et al., 2021; Chen et al., 2021; Kitamura et al., 2024). Work by Vaswani et al. (2022) have showed their algorithm uses no more than $\tilde{O}\left(\frac{SA}{(1-\gamma)^3\epsilon^2}\right)$ samples to achieve relaxed feasibility and $\tilde{O}\left(\frac{SA}{(1-\gamma)^5\zeta^2\epsilon^2}\right)$ samples to achieve strict feasibility. Here, the $\gamma \in [0,1)$ is the discount factor and $\zeta \in (0, \frac{1}{1-\gamma}]$ is the Slater's constant, which characterizes the size of the feasible region and hence the hardness of the CMDP. In their work, they have also provided a lower bound of

$\Omega\left(\frac{SA}{(1-\gamma)^5\zeta^2\epsilon^2}\right)$ on the sample complexity under strict feasibility. However, all the aforementioned results all scale polynomially with the cardinality of the state space.

For problems with large or possibly infinite state spaces, works by (Jain et al., 2022; Ding et al., 2021; Miryoosefi and Jin, 2022; Ghosh et al., 2024; Liu et al., 2022) have used linear function approximations to address the curse of dimensionality. All these works, except Jain et al. (2022); Liu et al. (2022), make the linear MDP assumption, where the transition function is linearly representable.

Under the generative model, for the infinite horizon discounted case, the online algorithm proposed in Jain et al. (2022) achieves a regret of $\tilde{O}(\sqrt{d}/\sqrt{K})$ with $\tilde{O}(\sqrt{d}/\sqrt{K})$ constraint violation, where $K$ is the number of iterations. Work by Liu et al. (2022) is able to achieve a faster $O(\ln(K)/K)$ convergence rate for both the reward suboptimality and constraint violation. For the online access setting under linear MDP assumption, Ding et al. (2021); Ghosh et al. (2024) achieve a regret of $\tilde{O}(\text{poly}(d)\,\text{poly}(H)\sqrt{T})$ with $\tilde{O}(\text{poly}(d)\,\text{poly}(H)\sqrt{T}))$ violations, where $T$ is the number of episodes and $H$ is the horizon term.

Miryoosefi and Jin (2022) presented an algorithm that achieves a sample complexity of $\tilde{O}\left(\frac{d^3H^6}{\epsilon^2}\right)$, where $d$ is the dimension of the feature space and $H$ is the horizon term in the finite horizon CMDP setting. In the more general setting under $q_\pi$-realizability, the best-known upper bounds are in the unconstrained MDP setting.

In the unconstrained MDP setting with access to a local-access model, early work by Kearns et al. (2002) have developed a tree-search style algorithms under this model, albeit in the tabular setting. Under $v^*$-realizability, Weisz et al. (2021) presented a planner that returns an $\epsilon$-optimal policy using $O((dH/\epsilon)^{|\mathcal{A}|})$ queries to the simulator. More works by (Yin et al., 2022; Hao et al., 2022; Weisz et al., 2022) have considered the local-access model with $q_\pi$-realizability assumption. Recent work by Weisz et al. (2022) have shown their algorithm can return a near-optimal policy that achieves a sample complexity of $\tilde{O}\left(\frac{d}{(1-\gamma)^4\epsilon^2}\right)$.

## 3 Problem formulation

**Constrained MDP**

We consider an infinite-horizon discounted CMDP $(\mathcal{S}, \mathcal{A}, P, r, c, \gamma, b, s_0)$ consisting a possibly countably infinite state space $\mathcal{S}$ with a finite set of actions $\mathcal{A}$, a reward function $r : \mathcal{S} \times \mathcal{A} \to [0, 1]$, a constraint function $c : \mathcal{S} \times \mathcal{A} \to [0, 1]$, a discount factor $\gamma \in [0, 1)$, a constraint threshold $b \geq 0$, and a fixed initial state $s_0$. Let $\mathcal{M}_1(X)$ denote the space of probability distributions supported on the set $X$. Then, the transition probability $P : \mathcal{S} \times \mathcal{A} \to \mathcal{M}_1(\mathcal{S})$.

Define a set of stationary randomized policies $\Pi_{\text{rand}}$, and a policy $\pi \in \Pi_{\text{rand}}$ maps states to probability distributions over the actions (i.e., $\pi : \mathcal{S} \to \mathcal{M}_1(\mathcal{A})$). Given a $\pi \in \Pi_{\text{rand}}$, the policy $\pi$ interacts with the CMDP starting from any state $s \in \mathcal{S}$ through discrete steps indexed by $t \in \mathbb{N}_0$, where $\mathbb{N}_0 = \{0, 1, 2, \dots\}$. This interaction generates a trajectory of $\{S_t, A_t\}_{t\in\mathbb{N}_0}$, where $S_0 = s, A_t \sim \pi(\cdot|S_t)$, and $S_{t+1} \sim P(\cdot|S_t, A_t)$. The reward action-value function is defined as $q_\pi^r(s, a) = \mathbb{E}\left[\sum_{t=0}^\infty \gamma^t r(S_t, A_t)|S_0 = s, A_0 = a\right]$. Similarly, the constraint action-value function is defined as $q_\pi^c(s, a) = \mathbb{E}\left[\sum_{t=0}^\infty \gamma^t c(S_t, A_t)|S_0 = s, A_0 = a\right]$. The reward state-value function $v_\pi^r(s) = \langle\pi(\cdot|s), q_\pi^r(s, \cdot)\rangle$, where $\langle\cdot, \cdot\rangle$ denotes the inner product over actions. Likewise, the constraint state-value function $v_\pi^c(s) = \langle\pi(\cdot|s), q_\pi^c(s, \cdot)\rangle$.

The objective of the CMDP is to find a policy $\pi$ that maximizes the state-value function $v_\pi^r$ starting from a given state $s_0$, while ensuring that the constraint $v_\pi^c(s_0) \geq b$ is satisfied:

$$\max_{\pi\in\Pi_{\text{rand}}} v_\pi^r(s_0) \quad s.t. \quad v_\pi^c(s_0) \geq b. \tag{1}$$

We assume the existence of a feasible solution to eq. (1) and let $\pi^*$ denote a solution to eq. (1). A quantity unique to CMDP is the Slater's constant, which is denoted as $\zeta = \max_\pi v_\pi^c(s_0) - b$. Slater's constant characterizes the size of the feasibility region, and hence the hardness of the problem.

Because the state space can be large or possibly infinite, we use linear function approximation to approximate the values of stationary randomized policies. Let $\phi : \mathcal{S} \times \mathcal{A} \to \mathbb{R}^d$ be a feature map. We assume that both $q_\pi^r$ and $q_\pi^c$ satisfy the following condition:

**Assumption 1** *($q_\pi$-realizability) There exists $B > 0$ and a misspecification error $\omega \geq 0$ such that for every $\pi \in \Pi_{rand}$, there exists a weight vector $w_\pi \in \mathbb{R}^d$, $\|w_\pi\|_2 \leq B$, and ensures $|q_\pi(s, a) - \langle w_\pi, \phi(s, a)\rangle| \leq \omega$ for all $(s, a) \in \mathcal{S} \times \mathcal{A}$.*

A **mixture policy** is defined as a policy randomly selected from a finite set of policies $\{\pi_0, \cdots, \pi_K\}$ and executed for all subsequent steps. For example, a mixture policy $\bar{\pi}_K$ is constructed by sampling a policy $\pi_k$ with probability $\frac{1}{K}$ and following it. The value function of such mixture policy for state $s \in \mathcal{S}$ is given by $v_{\bar{\pi}_K}(s) = \frac{1}{K} \sum_{k=0}^{K-1} v_{\pi_k}(s)$, where $v_{\pi_k}(s)$ is the value function of the individual policy $\pi_k$. Note that $\bar{\pi}_K$ is a non-stationary policy, and the set of non-stationary policies includes the set of stationary randomized policies $\Pi_{rand}$.

We assume access to a local access model, where the agent can query the simulator only for states that have been encountered during previous simulations. Our goal is to design an algorithm that outputs a near-optimal mixture policy $\bar{\pi}_K$, whose performance can be characterized in one of two ways.

For a given target error $\epsilon > 0$, the **relaxed feasibility** requires the returned policy $\bar{\pi}_K$ whose sub-optimality gap $v^r_{\pi^*}(s_0) - v^r_{\bar{\pi}_K}(s_0)$ is bounded by $\epsilon$, while allowing for a small constraint violation. Formally, we require $\bar{\pi}_K$ such that

$$v^r_{\pi^*}(s_0) - v^r_{\bar{\pi}_K}(s_0) \leq \epsilon \quad s.t \quad v^c_{\bar{\pi}_K}(s_0) \geq b - \epsilon.$$

On the other hand, **strict-feasibility** requires the returned policy $\bar{\pi}_K$ whose sub-optimality gap $v^r_{\pi^*}(s_0) - v^r_{\bar{\pi}_K}(s_0)$ is bounded by $\epsilon$ while not allowing any constraint violation. Formally, we require $\bar{\pi}_K$ such that

$$v^r_{\pi^*}(s_0) - v^r_{\bar{\pi}_K}(s_0) \leq \epsilon \quad s.t \quad v^c_{\bar{\pi}_K}(s_0) \geq b.$$

**Notations**

For any real number $a \in \mathbb{R}$, we let $\lfloor a \rfloor$ to denote the smallest integer $i$ such that $i \leq a$. For vector $x \in \mathbb{R}^d$, let $\|x\|_1 = \sum_i |x_i|$, $\|x\|_2 = \sqrt{\sum_i x_i^2}$, and $\|x\|_\infty = \max_i |x_i|$. For a positive definite matrix $A \in \mathbb{R}^{d \times d}$, the $\|x\|_A^2 = x^\top A x$. We let $\text{proj}_{[a_1, a_2]}(\lambda) = \arg\min_{p \in [a_1, a_2]} |\lambda - p|$, and $\text{trunc}_{[a_1, a_2]}(y) = \min\{\max\{y, a_1\}, a_2\}$. For any two positive numbers $a, b$, we write $a = O(b)$ if there exists an absolute constant $c > 0$ such that $a \leq cb$. We use the $\tilde{O}$ to hide any polylogarithmic terms.

# 4 Confident-NPG-CMDP, a local-access algorithm for CMDP

In this section, we introduce a primal-dual algorithm, which we call *Confident-NPG-CMDP* (see algorithm 1).

## 4.1 A primal-dual approach

We approach solving the CMDP problem by framing it as an equivalent saddle-point problem:

$$\max_\pi \min_{\lambda \geq 0} L(\pi, \lambda),$$

where $L : \Pi_{rand} \times \mathbb{R}_+ \to \mathbb{R}$ is the Lagrange function. For a policy $\pi \in \Pi_{rand}$ and a Lagrange multiplier $\lambda \in \mathbb{R}_+$, we have

$$L(\pi, \lambda) = v^r_\pi(s_0) + \lambda(v^c_\pi(s_0) - b).$$

Let $(\pi^*, \lambda^*)$ be a solution to this saddle-point problem. By an equivalence to a LP formulation and strong duality (Altman, 2021), $\pi^*$ is the policy that achieves the optimal value in the CMDP as defined in eq. (1). An optimal Lagrange multiplier $\lambda^* \in \arg\min_{\lambda \geq 0} L(\pi^*, \lambda)$, Therefore, solving eq. (1) is equivalent to finding a saddle-point of the Lagrange function.

A typical primal dual algorithm that finds the saddle-point will proceed in an iterative fashion alternating between a policy update using policy gradient and a dual variable update using mirror descent. The policy gradient is computed with respect to the primal value $q^p_{\pi_k, \lambda_k} = q^r_{\pi_k} + \lambda_k q^c_{\pi_k}$ and the mirror descent is computed with respect to the constraint value $v^c_{\pi_k}(s_0) = \langle \pi_k(\cdot|s_0), q^c_{\pi_k}(s_0, \cdot)\rangle$.

**Algorithm 1** Confident-NPG-CMDP

1: **Input:** $s_0$ (initial state), $\epsilon$ (target accuracy), $\delta \in (0, 1]$ (failure probability); $\gamma$ (discount factor)
2: **Initialize:**
3: Define $K, \eta_1, \eta_2, m$ according to Theorem 1 for relaxed-feasibility and Theorem 2 for strict-feasibility,
4: Set $L \leftarrow \lfloor \lfloor K \rfloor / (\lfloor m \rfloor + 1) \rfloor$.
5: For each iteration $k \in \{0, \ldots, \lfloor K \rfloor\} : \pi_k \leftarrow \text{Unif}(\mathcal{A}), \tilde{Q}_k^p(\cdot, \cdot) \leftarrow 0, \tilde{V}_k^c(\cdot) \leftarrow 0,$ and $\lambda_k \leftarrow 0$.
6: For each phase $l \in \{0, \ldots, L+1\} : \mathcal{C}_l \leftarrow (), D_l \leftarrow \{\}$

7: For $a \in \mathcal{A}$: if $(s_0, a) \notin \text{ActionCov}(\mathcal{C}_0)$, then append $(s_0, a)$ to $\mathcal{C}_0$ and set $\perp$ to $D_0[(s_0, a)]$ $\quad \triangleright$ see ActionCov defined in eq. (4)

8: **while** True **do** $\hfill \triangleright$ main loop
9: $\quad$ Let $\ell$ be the smallest integer s.t. $D_\ell[z'] = \perp$ for some $z' \in \mathcal{C}_\ell$
10: $\quad$ Let $z$ be the first state-action pair in $\mathcal{C}_\ell$ s.t. $D_\ell[z] = \perp$

11: $\quad$ If $\ell = L + 1$, then return $\bar{\pi}_K$

12: $\quad$ $k_\ell \leftarrow \ell \times (\lfloor m \rfloor + 1)$ $\hfill \triangleright$ iteration corresponding to phase $\ell$
13: $\quad$ $(result, discovered) \leftarrow \text{Gather-data}(\pi_{k_\ell}, \mathcal{C}_\ell, \alpha, z)$
14: $\quad$ **if** $discovered$ is True **then**
15: $\quad\quad$ Append $result$ to $\mathcal{C}_0$ and set $\perp$ to $D_0[result]$ $\hfill \triangleright result$ is a state-action pair
16: $\quad\quad$ Goto line 8

17: $\quad$ $D_\ell[z] \leftarrow result$

18: $\quad$ **if** $\nexists z' \in \mathcal{C}_\ell$ s.t. $D_\ell[z'] = \perp$ **then**
19: $\quad\quad$ $k_{\ell+1} \leftarrow k_\ell + (\lfloor m \rfloor + 1)$ if $k_\ell + (\lfloor m \rfloor + 1) \leq \lfloor K \rfloor$ otherwise $\lfloor K \rfloor$

20: $\quad\quad$ **for** $k = k_\ell, \ldots, k_{\ell+1} - 1$ **do** $\hfill \triangleright$ off-policy iterations reusing $\mathcal{C}_\ell, D_\ell$
21: $\quad\quad\quad$ $Q_k^r, Q_k^c \leftarrow LSE(\mathcal{C}_\ell, D_\ell, \pi_k, \pi_{k_\ell})$

22: $\quad\quad\quad$ For $s \in \text{Cov}(\mathcal{C}_\ell) \setminus Cov(\mathcal{C}_{\ell+1})$, and for $a \in \mathcal{A}$
23: $\quad\quad\quad$ $\tilde{Q}_k^p(s, a) \leftarrow \text{trunc}_{[0, \frac{1}{1-\gamma}]} Q_k^r(s, a) + \lambda_k \text{trunc}_{[0, \frac{1}{1-\gamma}]} Q_k^c(s, a)$

24: $\quad\quad\quad$ $\tilde{V}_k^c(s) \leftarrow \text{trunc}_{[0, \frac{1}{1-\gamma}]} \langle \pi_k(\cdot|s), Q_k^c(s, \cdot) \rangle$

25: $\quad\quad\quad$ $\triangleright$ update policy
26: $\quad\quad\quad$ For $s, a \in \mathcal{S} \times \mathcal{A}$:
27: $\quad\quad\quad$ $\pi_{k+1}(a|s) \leftarrow \begin{cases} \pi_{k+1}(a|s) & \text{if } s \in \text{Cov}(\mathcal{C}_{\ell+1}) \\ \pi_k(a|s) \frac{\exp(\eta_1 \tilde{Q}_k^p(s,a))}{\sum_{a' \in \mathcal{A}} \pi_k(a'|s) \exp(\eta_1 \tilde{Q}_k^p(s,a'))} & \text{otherwise} \end{cases}$

28: $\quad\quad\quad$ $\triangleright$ update dual variable
29: $\quad\quad\quad$ $\lambda_{k+1} \leftarrow \begin{cases} \lambda_{k+1} & \text{if } s_0 \in \text{Cov}(\mathcal{C}_{\ell+1}) \\ \text{proj}_{[0,U]} \left( \lambda_k - \eta_2 (\tilde{V}_k^c(s_0) - b) \right) & \text{otherwise.} \end{cases}$

30: $\quad\quad$ For $z \in \mathcal{C}_\ell$ s.t. $z \notin \mathcal{C}_{\ell+1}$: append $z$ to $\mathcal{C}_{\ell+1}$ and set $\perp$ to $D_{\ell+1}[z]$

Given that we do not have access to an oracle for exact policy evaluations, we must collect data to estimate the primal and constraint values. If we have the least-squares estimates of $q_{\pi_k}^r$ and $q_{\pi_k}^c$, denoted by $Q_k^r$ and $Q_k^c$, respectively, then we can compute the least-squares estimate $Q_k^p = Q_k^r + \lambda_k Q_k^c$ to be the estimate of the primal value $q_{\pi_k, \lambda_k}^p$. Additionally, we can compute $V_k^c(s_0) = \langle \pi_k(\cdot|s_0), Q_k^c(s_0, \cdot) \rangle$ to be the least-squares estimate of the constraint value $v_{\pi_k}^c(s_0)$. Then, for any given $(s, a) \in \mathcal{S} \times \mathcal{A}$, our algorithm makes a policy update of the following form:

$$\pi_{k+1}(a|s) \propto \pi_k(a|s) \exp(\eta_1 Q_k^p(s, a)), \tag{2}$$

followed by a dual variable update of the following form:

$$\lambda_{k+1} \leftarrow \lambda_k - \eta_2 \left( V_k^c(s_0) - b \right),$$

where the $\eta_1$ and $\eta_2$ are the step-sizes.

## 4.2 Core set and least square estimates

To construct the least-squares estimates, let us assume for now that we are given a set of state-action pairs, which we call the core set $\mathcal{C}$. By organizing the feature vector of each state-action pair in $\mathcal{C}$ row-wise into a matrix $\Phi_{\mathcal{C}} \in \mathbb{R}^{|\mathcal{C}| \times d}$, we can write the covariance matrix as $V(\mathcal{C}, \alpha) = \Phi_{\mathcal{C}}^\top \Phi_{\mathcal{C}} + \alpha I$. For each $(s, a) \in \mathcal{C}$, suppose we have run Monte Carlo rollouts using the rollout policy $\pi$ with the local access simulator to obtain an averaged Monte Carlo return denoted by $\bar{q}(s, a)$. Then we gather all the state-action pairs into a vector $\bar{q} \in \mathbb{R}^{|\mathcal{C}|}$. For any state-action pair $(s, a) \in \mathcal{S} \times \mathcal{A}$, the least-square estimate of action-value $q_\pi$ is defined to be

$$Q(s, a) = \langle \phi(s, a), V(\mathcal{C}, \alpha)^{-1} \Phi_{\mathcal{C}}^\top \bar{q} \rangle. \tag{3}$$

Since the algorithm can only rely on estimates for policy improvement and constraint evaluation, it is imperative that these estimates closely approximate their true action values. In the local access setting, an algorithm may not be able to visit all state-action pairs, so we cannot guarantee that the estimates will closely approximate the true action values for all state-action pairs. However, we can ensure the accuracy of the estimates for a subset of states.

Given $\mathcal{C}$, let us define a set of state-action pairs whose features satisfies the condition $\|\phi(s, a)\|_{V(\mathcal{C}, \alpha)^{-1}} \leq 1$, then we call this set the action-cover of $\mathcal{C}$:

$$\text{ActionCov}(\mathcal{C}) = \{(s, a) \in \mathcal{S} \times \mathcal{A} : \|\phi(s, a)\|_{V(\mathcal{C}, \alpha)^{-1}} \leq 1\}. \tag{4}$$

Following from the action-cover, we have the cover of $\mathcal{C}$. For a state $s$ to be in the cover of $\mathcal{C}$, all its actions $a \in \mathcal{A}$, the pair $(s, a)$ is in the action-cover of $\mathcal{C}$. In other words,

$$\text{Cov}(\mathcal{C}) = \{s \in \mathcal{S} : \forall a \in \mathcal{A}, (s, a) \in \text{ActionCov}(\mathcal{C})\}.$$

For any $s \in \text{Cov}(\mathcal{C})$, we can ensure the least square estimate $Q(s, a)$ defined by eq. (3) closely approximates its true action value $q_\pi(s, a)$ for all $a \in \mathcal{A}$. However, such a core set $\mathcal{C}$ is not available before the algorithm is run. Therefore, we need an algorithm that will build a core set incrementally in the local-access setting while planning. To achieve this, we build our algorithm on CAPI-QPI-Plan (Weisz et al., 2022), using similar methodology for core set building and data gathering.

## 4.3 Core set building and data gathering to control the accuracy of the least-square estimates

Confident-NPG-CMDP does not collect data in every iteration but collects data in interval of $m = O\left(\ln(1 + \rho_0) \text{poly}(\epsilon^{-1}(1 - \gamma)^{-1})\right)$, where $\rho_0 \geq 0$ is an user defined constant. During each data collection phase, the algorithm performs on-policy evaluation. Between these phases, it conducts $(\lfloor m \rfloor + 1)$ off-policy evaluations, reusing data from the most recent on-policy iteration.

By setting $\rho_0$ to a positive value, we impose an upper bound of $1 + \rho_0$ on the per-trajectory importance sampling ratio used in off-policy evaluations, and $m$ is adjusted accordingly to maintain this bound. The total number of data collection phases is $L = \lfloor \lfloor K \rfloor / (\lfloor m \rfloor + 1) \rfloor$, where $K$ is the total number of iterations. When $\rho_0$ is set to zero, we have $L = K$, resulting in a purely on-policy version of the algorithm.

Confident-NPG-CMDP maintains a set of core sets $\{\mathcal{C}_l\}_{l=0}^{L+1}$, one for each data collection phases. Each core set $\mathcal{C}_l$ is a list of state-action pairs. Due to the off-policy evaluations, Confident-NPG-CMDP also maintains a set of data sets $\{D_l\}_{l=0}^{L}$. Initially, all core sets are empty, all policies are initialized to the uniform policy, and all data sets are empty.

The algorithm begins by adding the feature vectors corresponding to $(s_0, a)$ for all actions $a \in \mathcal{A}$ that are not in the action-cover of $\mathcal{C}_0$. These feature vectors are considered informative. For every $(s, a) \in \mathcal{C}_0$, the algorithm adds an entry to $D_0$ and sets its value to the placeholder $\perp$, indicating that there is no roll-out data yet. Then, in line 9 of algorithm 1, the algorithm finds the smallest integer $l \in \{0, \ldots, L\}$ such that the corresponding $D_l$ has an entry without roll-out data (i.e., it contains the placeholder $\perp$). When such a phase is found, a running phase begins, denoted by $\ell$ in algorithm 1. We note that when $\ell = L + 1$, the algorithm returns and no roll-outs are stored.

Since only one running phase $\ell$ can be active at a time, and $\ell$ can only take value $l \in \{0, \ldots, L\}$, the algorithm updates the policies of the corresponding iterations in line 27, updates the dual variables of these iterations in line 29, and extends the core set for the next phase in line 30.

Suppose during a running phase with $\ell = l$, while performing the roll-out in Gather-data subroutine (algorithm 3 in Appendix A), if any state-action pair $(s, a) \in \mathcal{S} \times \mathcal{A}$ is not in the action-cover of $\mathcal{C}_\ell$, the current running phase stops and the newly discovered state-action pair is added to $\mathcal{C}_0$ in line 15. The same state-action pair is then propagated to $\mathcal{C}_1$ and so on by line 30.

Once a state-action pair is added to a core set by line 7, line 15, and line 30, it remains in that core set for the duration of the algorithm. This means that any $\mathcal{C}_l$, $l \in \{0, \ldots, L + 1\}$ can grow in size and be extended multiple times during the execution of the algorithm. When any new state-action pair is added to a core set, the least-square estimate should be recomputed with the newly added information. This implies that the policy needs to be updated and data re-collected. However, we can avoid restarting the entire data collection procedure by updating only the policy for states that are newly added to the extended core set. We elaborate on this approach further in the next paragraph.

When the algorithm enters the running phase $\ell = l$, and the Gather-data subroutine returns, the LSE subroutine (algorithm 4) computes the least-squares estimate $Q_k^r, Q_k^c$ using the most recently extended core set $\mathcal{C}_\ell$ for each corresponding iteration $k = k_\ell, \ldots, k_{\ell+1} - 1$. Subsequently, $\tilde{Q}_k^p$ of line 23 of algorithm 1 is updated with the newly updated least-square estimates $Q_k^r, Q_k^c$. However, the policy $\pi_{k+1}$ will only be updated for states that are newly covered by $\mathcal{C}_\ell$ (i.e., $s \in \mathrm{Cov}(\mathcal{C}_\ell) \setminus \mathrm{Cov}(\mathcal{C}_{\ell+1})$). For any states that are already covered by $\mathcal{C}_\ell$ (i.e., $s \in \mathrm{Cov}(\mathcal{C}_{\ell+1})$), the policy remains unchanged from its previous update using the $\tilde{Q}^p$ at that time. By updating the policy in this manner, the accuracy guarantee of $\tilde{Q}_k^p(s, a)$ with respect to $q_{\pi_k, \lambda_k}^p(s, a)$ is ensured not just for $\pi_k$, but for an extended set of policies defined as follows:

**Definition 1** *For any policy $\pi$ from the set of randomized policies $\Pi_{rand}$ and any subset $\mathcal{X} \subseteq \mathcal{S}$, the extended set of policies is defined as:*

$$\Pi_{\pi, \mathcal{X}} = \{\pi' \in \Pi_{rand} \mid \pi(\cdot|s) = \pi'(\cdot|s) \text{ for all } s \in \mathcal{X}\}.$$

By maintaining a set of core sets, gathering data via the Gather-data subroutine (algorithm 3 in Appendix A), making policy updates by line 27, and dual variable updates by line 29, we have:

**Lemma 1** *Whenever LSE subroutine in line 21 of Confident-NPG-CMDP is executed during a running phase $\ell = l$ for $l \in \{0, \ldots, L\}$, the least-square estimate $\tilde{Q}_k^p(s, a)$ satisfies the following condition for all iterations $k = k_\ell, \ldots, k_{\ell+1} - 1$ associated with this phase and for all $s \in \mathrm{Cov}(\mathcal{C}_\ell)$ and $a \in \mathcal{A}$,*

$$|\tilde{Q}_k^p(s, a) - q_{\pi_k', \lambda_k}^p(s, a)| \le \epsilon' \quad \text{for all } \pi_k' \in \Pi_{\pi_k, \mathrm{Cov}(\mathcal{C}_\ell)}, \tag{5}$$

*where $\epsilon' = (1 + U)(\omega + \sqrt{\alpha}B + (\omega + \epsilon)\sqrt{\tilde{d}})$ with $\tilde{d} = \tilde{O}(d)$ and $U$ is an upper bound on the optimal Lagrange multiplier. Similarly, for initial state $s_0$, we have*

$$|\tilde{V}_k^c(s_0) - v_{\pi_k'}^c(s_0)| \le \omega + \sqrt{\alpha}B + (\omega + \epsilon)\sqrt{\tilde{d}} \quad \text{for all } \pi_k' \in \Pi_{\pi_k, \mathrm{Cov}(\mathcal{C}_\ell)}. \tag{6}$$

The accuracy guarantee of eq. (5) and eq. (6) are maintained throughout the execution of the algorithm. By lemma 4.5 of Weisz et al. (2022) (restated in lemma 6 in Appendix A), for any past version $\mathcal{C}_l^{\mathrm{past}}$

of $\mathcal{C}_l$ and the corresponding policy $\pi_k^{\text{past}}$ associated with $\mathcal{C}_l^{\text{past}}$, we have $\Pi_{\pi_k, \text{Cov}(\mathcal{C}_l)} \subseteq \Pi_{\pi_k^{\text{past}}, \text{Cov}(\mathcal{C}_l^{\text{past}})}$. This means that if eq. (5) and eq. (6) hold true for any policy in $\Pi_{\pi_k^{\text{past}}, \text{Cov}(\mathcal{C}_l^{\text{past}})}$, they will also hold true for any future updated policy $\pi_k$.

## 4.4 Differences between Confident-NPG-CMDP and CAPI-QPI-Plan

CAPI-QPI-Plan is designed for unconstrained MDPs and returns a deterministic policy, which may not be feasible in the constrained setting. In contrast, Confident-NPG-CMDP returns a soft mixture policy $\bar{\pi}_K$, ensuring that $\bar{\pi}_K(a|s) > 0$ for all $(s, a) \in \mathcal{S} \times \mathcal{A}$.

In constrained MDPs, controlling the dual variable via mirror descent adds an $\epsilon^{-2}$ factor to the sample complexity. Directly applying CAPI-QPI-Plan would increase the complexity to $\tilde{O}(\epsilon^{-4})$ due to the need to manage both the dual variable and estimation error. To address this, Confident-NPG-CMDP employs the natural policy gradient for policy improvement and leverages the softmax policy structure to perform off-policy estimation, thereby reducing the complexity to $\tilde{O}(\epsilon^{-3})$.

By employing a per-trajectory importance sampling ratio, we weigh the Monte Carlo returns generated from data collected in earlier on-policy phases, resulting in unbiased estimates of action values with respect to the target policy. However, this ratio can become large if there is a substantial difference between the on-policy and target policies. To mitigate this, the algorithm collects data at intervals of $m$, effectively determining when to gather new data as the policy significantly diverges from an earlier recent data-gathering iteration. By setting $\rho_0 > 0$, we can bound the per-trajectory importance sampling ratio, thus controlling the interval $m$ for resampling on-policy data to produce well-controlled estimators.

Key algorithmic differences between Confident-NPG-CMDP and CAPI-QPI-Plan:

1. **Policy Improvement Step:** Confident-NPG-CMDP utilizes a softmax over the estimated action-values, whereas CAPI-QPI-Plan employs a greedy approach.

2. **Dual Variable Computation:** Confident-NPG-CMDP requires computation of the dual variable inherent in primal-dual algorithms.

3. **Data Sampling Strategy:** Unlike CAPI-QPI-Plan, Confident-NPG-CMDP does not sample data at every iteration but collects data at specific intervals to control the importance sampling ratio.

In the next two sections, we will demonstrate how these changes ensure a feasible mixture policy for the CMDP and address the additional analytical challenges.

# 5 Confident-NPG-CMDP satisfies relaxed-feasibility

With the accuracy guarantee of the least-square estimates, we prove that at the termination of Confident-NPG-CMDP, the returned mixture policy $\bar{\pi}_K$ satisfies relaxed-feasibility. We note that because of the execution of line 30 in algorithm 1, at termination, one can show using induction that $\mathcal{C}_0 = \mathcal{C}_1 = \cdots = \mathcal{C}_{L+1}$. Therefore, $\text{Cov}(\mathcal{C}_0) = \text{Cov}(\mathcal{C}_1) = \cdots = \text{Cov}(\mathcal{C}_L)$. Thus, it is sufficient to only consider $\mathcal{C}_0$ at the termination of the algorithm. By line 7 of algorithm 1, we have ensured $s_0 \in \text{Cov}(\mathcal{C}_0)$.

By employing the primal-dual approach discussed in section 4, we reduce the CMDP problem to an unconstrained problem with a single reward function of the form $r_\lambda = r + \lambda c$. Therefore, we can apply lemma 12 from the Confident-NPG algorithm in the single-reward setting (see Appendix A) to our Confident-NPG-CMDP algorithm, replacing $\pi$ with $\pi^*$. Consequently, the value difference between $\pi^*$ and $\bar{\pi}_K$ can be bounded, which leads to:

**Lemma 2** *Let $\delta \in (0, 1]$ be the failure probability, $\epsilon > 0$ be the target accuracy, and $s_0$ be the initial state. Assuming for all $s \in \text{Cov}(\mathcal{C}_0)$ and all $a \in \mathcal{A}$, $|\tilde{Q}_k^p(s, a) - q_{\pi_k', \lambda_k}^p(s, a)| \leq \epsilon'$ and $|\tilde{V}_k^c(s_0) - v_{\pi_k'}^c(s_0)| \leq \omega + \sqrt{\alpha}B + (\omega + \epsilon)\sqrt{\tilde{d}}$ for all $\pi_k' \in \Pi_{\pi_k, \text{Cov}(\mathcal{C}_0)}$, then, with probability $1 - \delta$,*

*Confident-NPG-CMDP returns a mixture policy $\bar{\pi}_K$ that satisfies the following,*

$$v_{\pi^*}^r(s_0) - v_{\bar{\pi}_K}^r(s_0) \leq \frac{5\epsilon'}{1-\gamma} + \frac{(\sqrt{2\ln(A)}+1)(1+U)}{(1-\gamma)^2\sqrt{K}},$$

$$b - v_{\bar{\pi}_K}^c(s_0) \leq [b - v_{\bar{\pi}_K}^c(s_0)]_+ \leq \frac{5\epsilon'}{(1-\gamma)(U-\lambda^*)} + \frac{(\sqrt{2\ln(A)}+1)(1+U)}{(1-\gamma)^2(U-\lambda^*)\sqrt{K}},$$

*where $\epsilon' = (1+U)(\omega + (\sqrt{\alpha}B + (\omega+\epsilon)\sqrt{\tilde{d}}))$ with $\tilde{d} = \tilde{O}(d)$, and $U$ is an upper bound on the optimal Lagrange multiplier.*

By setting the parameters to appropriate values, it follows from lemma 2 that we obtain the following result:

**Theorem 1** *With probability $1-\delta$, the mixture policy $\bar{\pi}_K$ returned by confident-NPG-CMDP ensures that*

$$v_{\pi^*}^r(s_0) - v_{\bar{\pi}_K}^r(s_0) = \tilde{O}(\sqrt{d}(1-\gamma)^{-2}\zeta^{-1}\omega) + \epsilon,$$

$$v_{\bar{\pi}_K}^c(s_0) \geq b - \left(\tilde{O}(\sqrt{d}(1-\gamma)^{-2}\zeta^{-1}\omega) + \epsilon\right).$$

*if we choose $n = \tilde{O}(\epsilon^{-2}\zeta^{-2}(1-\gamma)^{-4}d)$, $\alpha = O\left(\epsilon^2\zeta^2(1-\gamma)^4\right)$, $K = \tilde{O}\left(\epsilon^{-2}\zeta^{-2}(1-\gamma)^{-6}\right)$, $\eta_1 = \tilde{O}\left((1-\gamma)^2\zeta K^{-1/2}\right)$, $\eta_2 = \zeta^{-1}K^{-1/2}$, $H = \tilde{O}\left((1-\gamma)^{-1}\right)$, $m = \tilde{O}\left(\epsilon^{-1}\zeta^{-2}(1-\gamma)^{-2}\right)$, and $L = \lfloor K/(\lfloor m\rfloor + 1)\rfloor = \tilde{O}\left(\epsilon^{-1}(1-\gamma)^{-4}\right)$ total number of data collection phases.*

*Furthermore, the algorithm utilizes at most $\tilde{O}(\epsilon^{-3}\zeta^{-3}d^2(1-\gamma)^{-11})$ queries in the local-access setting.*

**Remark 1:** In the presence of misspecification error $\omega > 0$, the reward suboptimality and constraint violation is $\tilde{O}(\omega) + \epsilon$ with the same sample complexity.

**Remark 2:** Suppose the Slater's constant $\zeta$ is much smaller than the suboptimality bound of $\tilde{O}(\omega) + \epsilon$, and it is reasonable to set $\zeta = \epsilon$. Then, the sample complexity is $\tilde{O}(\epsilon^{-6}(1-\gamma)^{-11}d^2)$, which is independent of $\zeta$.

**Remark 3:** Our algorithm requires the Slater's constant $\zeta$, which can be estimated by running CAPI-QPI-Plan only on the constraint function $c$, treating it as an unconstrained optimization problem. This yields an approximation of $\max_\pi v_\pi^c(s_0)$, allowing us to estimate $\zeta$. Performing this estimation before executing Confident-NPG-CMDP adds only an additive term to the overall sample complexity.

## 6 Confident-NPG-CMDP satisfies strict-feasibility

To address the strict feasibility problem, where no constraint violations are permitted (i.e., $v_{\bar{\pi}_K}^c \geq b$), the algorithm must solve a more conservative CMDP. We define a surrogate CMDP with the tuple $(\mathcal{S}, \mathcal{A}, P, r, c, \gamma, b', s_0)$, where $b' = b + \Delta$ for some $\Delta \geq 0$. Note that $b' \geq b$, imposing stricter constraints than the original problem. The optimal policy of this surrogate CMDP ensures compliance with the original constraint and is defined as follows:

$$\pi_\triangle^* \in \arg\max v_\pi^r(s_0) \quad s.t. \quad v_\pi^c(s_0) \geq b'. \tag{7}$$

Notice that $\pi_\triangle^*$ is a more conservative policy than $\pi^*$, where $\pi^*$ is the optimal policy of the original CMDP objective eq. (1). By solving this surrogate CMDP using Confident-NPG-CMDP and applying the result of theorem 1, we obtain a $\bar{\pi}_K$ that would satisfy

$$v_{\pi^*}^r(s_0) - v_{\bar{\pi}_K}^r(s_0) \leq \bar{\epsilon} \quad s.t. \quad v_{\bar{\pi}_K}^c(s_0) \geq \quad b' - \bar{\epsilon},$$

where $\bar{\epsilon} = \tilde{O}(\omega) + \epsilon$. Expanding out $b'$, we have $v_{\bar{\pi}_K}^c(s_0) \geq b + \Delta - \bar{\epsilon}$. If we can set $\triangle$ such that $\triangle - \bar{\epsilon} \geq 0$, then $v_{\bar{\pi}_K}^c(s_0) \geq b$, which satisfies strict-feasibility. We show this formally in the next theorem, where $\triangle = O(\epsilon(1-\gamma)\zeta)$ and is incorporated into the algorithmic parameters for ease of presentation.

**Theorem 2** *With probability $1 - \delta$, a target $\epsilon > 0$, the mixture policy $\bar{\pi}_K$ returned by confident-NPG-CMDP ensures that $v_{\pi^*}^r(s_0) - v_{\bar{\pi}_K}^r(s_0) \leq \epsilon$ and $v_{\bar{\pi}_K}^c(s_0) \geq b$, if assuming the misspecification error $\omega \leq \epsilon \zeta^2 (1 - \gamma)^3 (1 + \sqrt{\tilde{d}})^{-1}$, and if we choose $\alpha = O\left(\epsilon^2 \zeta^3 (1 - \gamma)^5\right), K = \tilde{O}\left(\epsilon^{-2} \zeta^{-4} (1 - \gamma)^{-8}\right), n = \tilde{O}\left(\epsilon^{-2} \zeta^{-4} (1 - \gamma)^{-8} d\right), H = \tilde{O}\left((1 - \gamma)^{-1}\right), m = \tilde{O}\left(\epsilon^{-1} \zeta^{-2} (1 - \gamma)^{-3}\right), and L = \lfloor K/(\lfloor m \rfloor + 1) \rfloor = \tilde{O}((\epsilon^{-1} \zeta^{-2} (1 - \gamma)^{-5}))$ total data collection phases.*

*Furthermore, the algorithm utilizes at most $\tilde{O}(\epsilon^{-3} \zeta^{-6} (1 - \gamma)^{-14} d^2)$ queries in the local-access setting.*

**Remark 1:** We note that by solving this conservative CMDP incurs a higher sample complexity, necessitating a separate treatment for this setting. Additionally, in the presence of a misspecification error $\omega > 0$, the strict-feasibility setting requires additional assumptions on $\omega$, whereas the relaxed-feasibility setting does not. The sample complexity of the relaxed-feasibility setting can be independent of Slater's constant, whereas for strict feasibility, the returned policy must strictly adhere to constraints, and we cannot simply set Slater's constant $\zeta$ to $\epsilon$ and disregard its impact.

## 7 A discussion on memory cost and some implementation details

The overall memory requirement is $\tilde{d}nH(L + 1) + \tilde{d} + (L + 1)(m + 1)\tilde{d}d$. The term $\tilde{d}nH(L + 1)$ comes from maintaining $L + 1$ copies of the core sets, and each core set contains no more than $\tilde{d}$ state-action pairs. For each state-action pair in $\mathcal{C}_l$ for $l \in \{0, \dots, L\}$, the algorithm stores $n$ trajectories consisting of $H$ tuples $(s, a, r, c)$.

In phase $L + 1$, the algorithm terminates, so no roll-outs are stored. The second term $\tilde{d}$ accounts for the elements stored in $\mathcal{C}_{L+1}$, which has no more than $\tilde{d}$ elements.

Finally, the last term is the memory required to store the least-square weights of the estimator during core set extensions. Each core set $\mathcal{C}_l$ can undergo up to $\tilde{d}$ extensions. Recall that one state-action pair is added to $\mathcal{C}_0$ at a time, and subsequently propagated to $\mathcal{C}_1, \mathcal{C}_2$ and so on, ensuring that each core set contains no more than $\tilde{d}$ elements. During every extension of $\mathcal{C}_l$, the newly added state-action pairs are marked, and up to $m + 1$ least-square weights are stored to account for the corresponding iterations associated with $\mathcal{C}_l$. Since each weight vector has a dimension $d$, and there are $L + 1$ core sets maintained this manner, the total memory required to store all least-square weights is bounded by $(L + 1)(m + 1)\tilde{d}d$.

We store the least-squares weights because the algorithm must return a mixture policy, which requires access to all policies $\pi_0, \dots, \pi_{K-1}$. Instead of storing each $\pi_k$ for $k = 0, \dots, K - 1$ across the entire state-action space, the algorithm tracks the state-action pairs newly added to each core set during extensions and saves their corresponding least-squares weights for each extension. With this stored information and the initialization of $\pi_0$, a subroutine can reconstruct the policies $\pi_k(\cdot|s)$ for any $s$ and iteration $k$ as needed. Please refer to Appendix E for a brief discussion on how to mark the state-action pairs, store the least-square weights, and use this information to reconstruct the policies as required.

## 8 Conclusion

We have presented a primal-dual algorithm for planning in CMDP with large state spaces, given $q_\pi$-realizable function approximation. The algorithm, with high probability, returns a policy that achieves both the relaxed and strict feasibility CMDP objectives, using no more than $\tilde{O}(\epsilon^{-3}d^2 \operatorname{poly}(\zeta^{-1}(1 - \gamma)^{-1}))$ queries to the local-access simulator.

Our algorithm does not query the simulator and collect data in every iteration. Instead, the algorithm queries the simulator only at fixed intervals. Between these data collection intervals, our algorithm improves the policy using off-policy optimization. This approach makes it possible to achieve the desired sample complexity in both feasibility settings.

## Acknowledgments and Disclosure of Funding

Tian Tian would like to thank Roshan Shariff and Kenny Young for their insightful comments and helpful feedback during the preparation of this manuscript. Csaba Szepesvári also gratefully acknowledges funding from the Canada CIFAR AI Chairs Program, Amii, and NSERC. Lin Yang is supported in part by NSF #2221871 and an Amazon Faculty Award.

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

# A  Confident-NPG in a single reward setting

The pseudo code of Confident-NPG with a single reward setting is the same as Confident-NPG-CMDP in algorithm 1, except that line 24 and line 29 will not appear in Confident-NPG. Additionally, the LSE subroutine returns just $Q^r$, and the policy update will be with respect to $\tilde{Q}^r$. For the complete pseudo code of Confident-NPG in the single reward setting, please see algorithm 2. In the following analysis, for convenience, we omit the superscript $r$.

---

**Algorithm 2** Confident-NPG

---

**Input:** $s_0$ (initial state), $\epsilon$ (target accuracy), $\delta \in (0,1]$ (failure probability), $c \geq 0$, $\gamma$, $K = \frac{2\ln(|\mathcal{A}|)}{(1-\gamma)^4\epsilon^2}$,
$\eta_1 = (1-\gamma)\sqrt{\frac{2\ln(|\mathcal{A}|)}{K}}$, $m = \frac{\ln(1+\rho_0)}{2\epsilon(1-\gamma)\ln\left(\frac{4}{\epsilon(1-\gamma)^2}\right)}$, $L = \lfloor \lfloor K \rfloor /(\lfloor m \rfloor + 1)\rfloor$.

**Initialize:** for each iteration $k \in \{0, \dots, \lfloor K \rfloor\} : \pi_k \leftarrow \text{Uniform}(\mathcal{A})$, $\tilde{Q}_k^r(s,a) \leftarrow 0$, for all $s, a \in \mathcal{S} \times \mathcal{A}$, and $\lambda_k \leftarrow 0$. For each phase $l \in \{0, \dots, L+1\} : \mathcal{C}_l \leftarrow ()$, $D_l \leftarrow \{\}$

1: **for** $a \in \mathcal{A}$ **do**
2:     **if** $(s_0, a) \notin \text{ActionCov}(\mathcal{C}_0)$ **then**
3:         Append $(s_0, a)$ to $\mathcal{C}_0$; $D_0[(s_0, a)] \leftarrow \perp$

4: **while** True **do**                                                          $\triangleright$ main loop
5:     Get the smallest integer $\ell$ s.t. $D_\ell[z'] = \perp$ for some $z' \in \mathcal{C}_\ell$
6:     Get the first state-action pair $z$ in $\mathcal{C}_\ell$ s.t. $D_\ell[z] = \perp$
7:     **if** $\ell = L+1$ **then return** $\bar{\pi}_K$

8:     $k_\ell \leftarrow \ell \times (\lfloor m \rfloor + 1)$                               $\triangleright$ iteration corresponding to phase $\ell$
9:     $(result, discovered) \leftarrow$ Gather-data$(\pi_{k_\ell}, \mathcal{C}_\ell, \alpha, z)$
10:     **if** $discovered$ is True **then**
11:         $\triangleright result$ is a state-action pair
12:         Append $result$ to $\mathcal{C}_0$; $D_0[result] \leftarrow \perp$
13:         **break**

14:     $\triangleright result$ is a set of $n$ $H$-horizon trajectories $\sim \pi_{k_\ell}$ starting at $z$
15:     $D_\ell[z] \leftarrow result$

16:     **if** $\nexists z' \in \mathcal{C}_\ell$ s.t. $D_\ell[z'] = \perp$ **then**
17:         $k_{\ell+1} \leftarrow k_\ell + (\lfloor m \rfloor + 1)$ if $k_\ell + (\lfloor m \rfloor + 1) \leq \lfloor K \rfloor$ otherwise $\lfloor K \rfloor$

18:         $\triangleright$ update policy for every $k \in [k_\ell, k_{\ell+1} - 1]$ using $\mathcal{C}_\ell, D_\ell$
19:         **for** $k = k_\ell, \dots, k_{\ell+1} - 1$ **do**
20:             $Q_k^r, \_ \leftarrow LSE(\mathcal{C}_\ell, D_\ell, \pi_k, \pi_{k_\ell})$

21:         $\triangleright$ update variables and improve policy
22:         **for all** $s \in \text{Cov}(\mathcal{C}_\ell) \setminus \text{Cov}(\mathcal{C}_{\ell+1})$, and for all $a \in \mathcal{A}$ **do**
23:             $\tilde{Q}_k^r(s,a) \leftarrow Trunc_{[0,\frac{1}{1-\gamma}]} Q_k^r(s,a)$
24:         **for all** $s, a \in \mathcal{S} \times \mathcal{A}$ **do**
25:             $\pi_{k+1}(a|s) \leftarrow \begin{cases} \pi_{k+1}(a|s) & \text{if } s \in \text{Cov}(\mathcal{C}_{\ell+1}) \\ \pi_k(a|s)\frac{\exp(\eta_1 \tilde{Q}_k^r(s,a))}{\sum_{a' \in \mathcal{A}} \pi_k(a'|s)\exp(\eta_1 \tilde{Q}_k^r(s,a'))} & \text{otherwise} \end{cases}$
26:         **for** $z \in \mathcal{C}_\ell$ s.t. $z \notin \mathcal{C}_{\ell+1}$ **do**
27:             Append $z$ to $\mathcal{C}_{\ell+1}$; $D_{\ell+1}[z] \leftarrow \perp$

---

## A.1  The Gather-data subroutine

Given a core set $\mathcal{C}$, a behaviour policy $\mu$, a starting state-action pair $(s,a) \in \mathcal{S} \times \mathcal{A}$ along with some algorithmic parameters, the Gather-data subroutine (algorithm 3) will either 1) return a newly

---

**Algorithm 3** Gather-data

---

**Input:** policy $\pi$, core set $\mathcal{C}$, regression parameter $\alpha = \frac{\epsilon^2(1-\gamma)^2}{25B^2(1+U)}$, $H = \frac{\ln(4/(\epsilon(1-\gamma)))}{1-\gamma}$, $n = \frac{(1+\rho_0)^2 \ln\left(\frac{8\bar{d}L}{\delta}\right)}{2\epsilon^2(1-\gamma)^2}$, starting state and action $(s, a)$

**Initialize:** $Trajectories \leftarrow ()$

1: **if** $s \notin \mathrm{Cov}(\mathcal{C})$ **then**
2:     **for** $a \in \mathcal{A}$ **do**
3:         **if** $(s, a) \notin ActionCov(\mathcal{C})$ **then** return $((s, a), True)$
4: **for** $i \in [n]$ **do**
5:     $\tau_{s,a}^i \leftarrow ()$
6:     $S_0^i \leftarrow s, \; A_0^i \leftarrow a$
7:     append $S_0^i, A_0^i$ to $\tau_{s,a}^i$

8:     **for** $h = 0, \dots, H-1$ **do**
9:         $S_{h+1}^i, R_{h+1}^i, C_{h+1}^i \leftarrow \mathrm{simulator}(S_h^i, A_h^i)$

10:         **if** $S_{h+1}^i \notin \mathrm{Cov}(\mathcal{C})$ **then**
11:             **for** $a \in \mathcal{A}$ **do**
12:                 **if** $(S_{h+1}^i, a) \notin ActionCov(\mathcal{C})$ **then** return $((S_{h+1}^i, a), True)$

13:         ▷ no new informative features discovered
14:         $A_{h+1}^i \sim \pi(\cdot|S_{h+1}^i)$
15:         append $R_{h+1}^i, C_{h+1}^i, S_{h+1}^i, A_{h+1}^i$ to $\tau_{s,a}^i$

16:     append $\tau_{s,a}^i$ to $Trajectories$
    **return** $(Trajectories, False)$

---

discovered state-action pair, or 2) return a set of $n$ trajectories. Each trajectory is generated by running the behaviour policy $\mu$ with the simulator for $H$ consecutive steps. For $i = 1, \dots, n$, let $\tau_{s,a}^i$ denote the $i$th trajectory starting from $s, a$ to be $\{S_0^i = s, A_0^i = a, R_1^i, C_1^i, \cdots, S_{H-1}^i, A_{H-1}^i, R_H^i, C_H^i, S_H^i\}$. Then the $i$-th discounted cumulative rewards $G(\tau_{s,a}^i) = \sum_{h=0}^{H-1} \gamma^h R_{h+1}^i$. For a target policy $\pi$, then the empirical mean of the discounted sum of rewards is as follows,

$$\bar{q}(s, a) = \frac{1}{n} \sum_{i=1}^{n} \rho(\tau_{s,a}^i) G(\tau_{s,a}^i), \tag{8}$$

where $\rho(\tau_{s,a}^i) = \Pi_{h=1}^{H-1} \frac{\pi(A_h^i|S_h^i)}{\mu(A_h^i|S_h^i)}$ is the per-trajectory importance sampling ratio.

For some given $\bar{s}$ and $\bar{a}$, we establish the following relationship between the target policy $\pi$ and the behavior policy $\mu$:

$$\pi(\bar{a}|\bar{s}) \propto \mu(\bar{a}|\bar{s}) \exp(f(\bar{s}, \bar{a})) \quad \text{s.t.} \quad \sup_{\bar{s}, \bar{a}} |f(\bar{s}, \bar{a})| \leq \frac{\ln(1 + \rho_0)}{2H}, \tag{9}$$

where $f(\bar{s}, \bar{a}) : \mathcal{S} \times \mathcal{A} \to \mathbb{R}^+$ and $\rho_0 \geq 0$ is a given constant. By establishing the relationship stated in eq. (9), the importance sampling ratio $\rho(\tau_{s,a}^i)$ can be bounded by $1 + \rho_0$ as this is proven in the following lemma:

**Lemma 3** *Suppose the trajectory* $\tau = (S_0, A_0, R_1, S_1, A_1, \cdots, S_{H-1}, A_{H-1}, R_H)$ *is sampled from a behaviour policy* $\mu$, *and* $\mu$ *is related to the target policy* $\pi$ *via eq.* (9). *The per-trajectory importance sampling ratio*

$$\rho(\tau) = \Pi_{h=1}^{H-1} \frac{\pi(A_h|S_h)}{\mu(A_h|S_h)} \leq 1 + \rho_0.$$

Proof: Let $l_0 = \sup_{s,a} |f(s,a)|$, where $f$ is defined in eq. (9). For any $(s,a) \in \{(S_h, A_h)\}_{h=1}^{H-1}$,

$$\pi(a|s) = \mu(a|s) \frac{\exp(f(s,a))}{\sum_{a'} \mu(a'|s) \exp(f(s,a'))} \le \mu(a|s) \frac{\exp(l_0)}{\sum_{a'} \mu(a'|s) \exp(-l_0)}$$
$$\le \mu(a|s) \exp(2l_0).$$

We see that $\frac{\pi(a|s)}{\mu(a|s)} \le \exp(2l_0)$, and it follows that $\Pi_{h=1}^{H-1} \frac{\pi(A_h|S_h)}{\mu(A_h|S_h)} \le \exp(2l_0 H)$. By assumption, $l_0 \le \frac{\ln(1+\rho_0)}{2H}$, then $\exp(2l_0 H) \le \exp\left(2H \frac{\ln(1+\rho_0)}{2H}\right) \le 1 + \rho_0$. ∎

Define the probability distribution $P_{\pi,s,a}$ over trajectory $\{S_h, A_h, R_{h+1}\}_{h \ge 0}$ as follows: the initial state $S_0$ is set deterministically to $s$, and the initial action $A_0$ is set deterministically to $a$. For each subsequent time step $h \ge 0$, the next state $S_{h+1}$ is sampled according to the transition probability $P(\cdot|S_h, A_h)$, and the next action $A_{h+1}$ is sampled from the policy $\pi(\cdot|S_{h+1})$. It follows that $\mathbb{E}_{\pi,s,a}$ denotes the expectation with respect to distribution $P_{\pi,s,a}$.

Now, we show that for all $(s,a) \in \mathcal{C}$, $|\bar{q}(s,a) - q_\pi(s,a)| \le \epsilon$, where $\epsilon > 0$ is a given target error. Additionally, the accuracy guarantee of $|\bar{q}(s,a) - q_\pi(s,a)| \le \epsilon$ continues to holds for the extended set of policies defined in definition 1. Formally, we state the main result of this section.

**Lemma 4** *For any $s, a \in \mathcal{S} \times \mathcal{A}, \mathcal{X} \subset \mathcal{S}$, the Gather-data subroutine will either return with $((s', a'), \text{True})$ for some $s' \notin \mathcal{X}$, or it will return with $(D[(s,a)], \text{False})$, where $D[(s,a)]$ is a set of $n$ independent trajectories generated by a behavior policy $\mu$ starting from $(s, a)$. When Gather-data returns False for $(s,a)$, we assume 1) the behavior policy $\mu$ and target policy $\pi$ for all the states and actions encountered in the trajectories stored in $D[(s,a)]$ satisfy eq. (9) and 2) $\bar{q}(s,a)$ is an unbiased estimate of $\mathbb{E}_{\pi',s,a}[\sum_{h=0}^{H-1} \gamma^h R_{h+1}]$ for all $\pi' \in \Pi_{\pi,\mathcal{X}}$. Then, the importance-weighted return $\bar{q}(s,a)$ constructed from $D[(s,a)]$ according to eq. (8) will, with probability $1 - \delta'$,*

$$|\bar{q}(s,a) - q_{\pi'}(s,a)| \le \epsilon \quad \text{for all } \pi' \in \Pi_{\pi,\mathcal{X}}.$$

Proof: The proof follows similar reasoning to Lemma 4.2 Weisz et al. (2022).

Recall $D[(s,a)]$ stores $n$ number of trajectories indexed by $i$, where each trajectory $\tau_{s,a}^i = (S_0^i = s, A_0^i = a, R_0^i, \ldots, S_{H-1}^i) \sim \mu$. The per-trajectory importance sampling ratio $\rho(\tau_{s,a}^i) = \Pi_{h=1}^{H-1} \frac{\pi(A_h^i|S_h^i)}{\mu(A_h^i|S_h^i)}$, and the discounted cumulative return is $\sum_{h=0}^{H-1} \gamma^h R_{h+1}^i$. By the triangle inequality,

$$|\bar{q}(s,a) - q_\pi(s,a)| = |\frac{1}{n} \sum_{i=1}^n \Pi_{h=0}^{H-1} \frac{\pi(A_h^i|S_h^i)}{\mu(A_h^i|S_h^i)} \sum_{h=0}^{H-1} \gamma^h R_{h+1}^i - q_\pi(s,a)|$$

$$\le |\frac{1}{n} \sum_{i=1}^n \rho(\tau_{s,a}^i) \sum_{h=0}^{H-1} \gamma^h R_{h+1}^i - \mathbb{E}_{\pi,s,a} \sum_{h=0}^{H-1} \gamma^h R_{h+1}| + |E_{\pi,s,a} \sum_{h=0}^{H-1} \gamma^h R_{h+1} - q_\pi(s,a)|. \quad (10)$$

The goal is to bound each of the two terms in eq. (10) by $\frac{\epsilon}{4}$ so that the sum of the two is $\frac{\epsilon}{2}$.

By assumption, the policies $\pi$ and $\mu$ satisfies eq. (9) for all state-action pairs $(S_h^i, A_h^i)$ extracted from the $i$-trajectory $\tau_{s,a}^i$. Second, $\bar{q}(s,a)$ is assumed to be an unbiased estimate of $\mathbb{E}_{\pi,s,a}\left[\sum_{h=0}^{H-1} \gamma^h R_{h+1}\right]$. Note that for all $i = 1, \ldots, n$, the importance weighted cumulative return $\rho(\tau_{s,a}^i) \sum_{h=0}^{H-1} \gamma^h R_{h+1}^i$ are independent random variables, and the value of each such random variable $\in \left[0, \frac{1+\rho_0}{1-\gamma}\right]$. This is because 1) $\sum_{h=0}^{H-1} \gamma^h R_{h+1}^i \le \frac{1}{1-\gamma}$ since the rewards take values in the range of $[0, 1]$, and 2) $\rho(\tau_{s,a}^i) \le 1 + \rho_0$ by lemma 3. We apply Hoeffding's inequality,

$$\mathbb{P}\left(\left|\frac{1}{n} \sum_{i=1}^n \rho(\tau_{s,a}^i) \sum_{h=0}^{H-1} \gamma^h R_{h+1}^i - \mathbb{E}_{\pi,s,a} \sum_{h=0}^{H-1} \gamma^h R_{h+1}\right| > \frac{\epsilon}{4}\right) \le 2 \exp\left(-\frac{2n\left(\frac{\epsilon}{4}\right)^2}{\left(\frac{1+\rho_0}{1-\gamma}\right)^2}\right).$$

Then, we have with probability $1 - \delta'/2$, where $\delta' = 2\exp\left(-\frac{2n\epsilon^2}{16\left(\frac{1+\rho_0}{1-\gamma}\right)^2}\right)$, the first term in eq. (10)
$|\frac{1}{n}\sum_{i=1}^n \rho(\tau_{s,a}^i)\sum_{h=0}^{H-1}\gamma^h R_{h+1}^i - \mathbb{E}_{\pi,s,a}\sum_{h=0}^{H-1}\gamma^h R_{h+1}| \le \frac{\epsilon}{4}$. For the second term in eq. (10),

$$|E_{\pi,s,a}\sum_{h=0}^{H-1}\gamma^h R_{h+1} - q_\pi(s,a)| = |\mathbb{E}_{\pi,s,a}\sum_{h=H}^{\infty}\gamma^h R_{h+1}| \le \frac{\gamma^H}{1-\gamma}.$$

By the choice of $H = \frac{\ln(4/(1-\gamma)\epsilon)}{1-\gamma}$, we have $\frac{\gamma^H}{1-\gamma} \le \frac{\epsilon}{4}$. Putting everything together, we get $|\bar{q}(s,a) - q_\pi(s,a)| \le \frac{\epsilon}{2}$. To get the final result, we need to upper bound $|q_\pi(s,a) - q_{\pi'}(s,a)|$ by $\frac{\epsilon}{2}$, so that $|\bar{q}(s,a) - q_{\pi'}(s,a)| \le |\bar{q}(s,a) - q_\pi(s,a)| + |q_\pi(s,a) - q_{\pi'}(s,a)| \le \epsilon$.

Recall that $\pi$ and $\pi'$ differs in distributions over states that are not in $\mathcal{X}$. For a trajectory $(S_0 = s, A_0 = a, S_1, \dots)$, let $T$ be the smallest positive integer such that $S_T \notin \mathcal{X}$, then the distribution of the trajectory $(S_0 = s, A_0 = a, S_1, \dots, S_T)$ are the same under $P_{\pi,s,a}$ and $P_{\pi',s,a}$ because $\pi(\cdot|s) = \pi'(\cdot|s)$ for all $s \in \mathcal{X}$. Then,

$$|q_\pi(s,a) - q_{\pi'}(s,a)| = \left|\mathbb{E}_{\pi,s,a}\left[\sum_{t=0}^{T-1}\gamma^t R_t + \gamma^T v_\pi(S_T)\right] - \mathbb{E}_{\pi',s,a}\left[\sum_{t=0}^{T-1}\gamma^t R_t + \gamma^T v_{\pi'}(S_T)\right]\right|$$

$$= \left|\mathbb{E}_{\pi,s,a}\left[\gamma^T v_\pi(S_T)\right] - \mathbb{E}_{\pi',s,a}\left[\gamma^T v_{\pi'}(S_T)\right]\right|$$

$$= \sum_{s'\in\mathcal{X},a'} P_{\pi,s,a}(S_{T-1} = s', A_{T-1} = a')P(S_T|s',a')\gamma^T v_\pi(S_T)$$

$$- \sum_{s'\in\mathcal{X},a'} P_{\pi',s,a}(S_{T-1} = s', A_{T-1} = a')P(S_T|s',a')\gamma^T v_{\pi'}(S_T)$$

$$= \sum_{s'\in\mathcal{X},a'} P_{\pi,s,a}(S_{T-1} = s', A_{T-1} = a')P(S_T|s',a')\gamma^T (v_\pi(S_T) - v_{\pi'}(S_T))$$

$$\le \frac{1}{1-\gamma}\sum_{s'\in\mathcal{X},a'} P_{\pi,s,a}(S_{T-1} = s', A_{T-1} = a')P(S_T|s',a')\gamma^T = \frac{1}{1-\gamma}\mathbb{E}_{\pi,s,a}[\gamma^T]$$

$$= \frac{1}{1-\gamma}\sum_{t=1}^{\infty}\gamma^t P_{\pi,s,a}(T = t) \le \frac{1}{1-\gamma}\sum_{t=1}^{H-1}P_{\pi,s,a}(T = t)\gamma^0 + \frac{1}{1-\gamma}\sum_{t=H}^{\infty}P_{\pi,s,a}(T \ge H)\gamma^H$$

$$\le \frac{1}{1-\gamma}P_{\pi,s,a}(1 \le T < H) + \frac{\gamma^H}{1-\gamma}.$$

Recall $P_{\pi,s,a}(1 \le T < H) = \sum_{t=1}^{H-1}\sum_{s'\in\mathcal{X},a'\in\mathcal{A}} P_{\pi,s,a}(S_{t-1} = s', A_{t-1} = a')P(S_T|s',a')$, and recall $S_0 = s, A_0 = a$, then by the law of total probability, we have

$P_{\pi,s,a}(S_t = s', A_t = a')$

$= \sum_{\substack{s_1,\dots,s_{t-1},\\a_1,\dots,a_{t-1}}} \Pi_{i=0}^{t-1}P(S_{i+1} = s_{i+1}|S_i = s_i, A_i = a_i)\left(\Pi_{i=1}^t\pi(A_i = a_i|S_i = s_i)\right)$

$= \sum_{\substack{s_1,\dots,s_{t-1},\\a_1,\dots,a_{t-1}}} \Pi_{i=0}^{t-1}P(S_{i+1} = s_{i+1}|S_i = s_i, A_i = a_i)\Pi_{i=1}^t\frac{\pi(A_i = a_i|S_i = s_i)}{\mu(A_i = a_i|S_i = s_i)}\mu(A_i = a_i|S_i = s_i)$

$\le (1 + \rho_0) \sum_{\substack{s_1,\dots,s_{t-1},\\a_1,\dots,a_{t-1}}} \Pi_{i=0}^{t-1}P(S_{i+1} = s_{i+1}|S_i = s_i, A_i = a_i)\Pi_{i=1}^t\mu(A_i = a_i|S_i = s_i)$   (11)

$= (1 + \rho_0)P_{\mu,s,a}(S_t = s', A_t = a').$

To get eq. (11), we use lemma 3 and that $1 \le t \le H - 1$.

Altogether, we have

$$|q_\pi(s,a) - q_{\pi'}(s,a)|$$

$$\leq \frac{1}{1-\gamma} \sum_{t=1}^{H-1} \sum_{s' \in \mathcal{X}, a' \in \mathcal{A}} (1+\rho_0) P_{\mu,s,a}(S_{t-1} = s', A_{t-1} = a') P(S_T | s', a')$$

$$+ \frac{\gamma^H}{1-\gamma}$$

$$\leq \frac{1+\rho_0}{1-\gamma} P_{\mu,s,a}(1 \leq T < H) + \frac{\epsilon}{4}.$$

Now, we bound $P_{\mu,s,a}(1 \leq T < H)$. For each $(s,a) \in \mathcal{X}$, in the $i$th rollouts, let $I_i(s,a)$ be an indicator function, where it takes the value 1 when the event that $S_T \notin \mathcal{X}$ occurs during $1 \leq T < H$. Then $\mathbb{E}_{\mu,s,a}[I_i(s,a)] = P_{\mu,s,a}(1 \leq T < H)$. By another Hoeffding's inequality,

$$\mathbb{P}\left( |\mathbb{E}_{\mu,s,a}[I_i(s,a)] - \frac{1}{n} \sum_{i=1}^n I_i(s,a)| > \frac{\epsilon(1-\gamma)}{4(1+\rho_0)} \right) \leq 2 \exp\left( -\frac{2n(\epsilon(1-\gamma)/4(1+\rho_0))^2}{(1)^2} \right)$$

$$= 2 \exp\left( -\frac{2n\epsilon^2}{16\left(\frac{1+\rho_0}{1-\gamma}\right)^2} \right) = \delta'.$$

Then, with probability $1 - \delta'/2$,

$$|\mathbb{E}_{\mu,s,a}[I_i(s,a)] - \frac{1}{n} \sum_{i=1}^n I_i(s,a)| \leq \frac{\epsilon(1-\gamma)}{4(1+\rho_0)}, \tag{12}$$

When Gather-data subroutine returns, all indicators $I_i(s,a) = 0$ for all $(s,a) \in \mathcal{X}$ and $i \in [n]$, then we have

$$P_{\mu,s,a}(1 \leq T < H) \leq \frac{\epsilon(1-\gamma)}{4(1+\rho_0)}. \tag{13}$$

Putting everything together, we have the result. ∎

## A.2 The LSE subroutine

---

**Algorithm 4** LSE

---

**Input:** $\mathcal{C}, D, \pi_k, \pi'_k$

1: **for** $s, a \in \mathcal{C}$ **do**
2:     **for** every $\tau_{s,a}^i \in D[(s,a)]$ for every $i \in [n]$ **do**
3:         extract $\{S_0^i, A_0^i, R_1^i, C_1^i, S_1^i, A_1^i \cdots S_H^i, A_H^i\}$ from $\tau_{s,a}^i$
4:         compute $G_r^i(s,a) \leftarrow \sum_{h=0}^{H-1} \gamma^h R_{h+1}^i$; $G_c^i(s,a) \leftarrow \sum_{h=0}^{H-1} \gamma^h C_{h+1}^i$
5:         compute $\rho^i(s,a) \leftarrow \Pi_{h=1}^{H-1} \frac{\pi_k(A_h^i | S_h^i)}{\pi'_k(A_h^i | S_h^i)}$
6:     $\bar{q}^r(s,a) \leftarrow \frac{1}{n} \sum_{i=1}^n \rho^i(s,a) G_r^i(s,a)$; $\bar{q}^c(s,a) \leftarrow \frac{1}{n} \sum_{i=1}^n \rho^i(s,a) G_c^i(s,a)$

7: $w^r \leftarrow \left(\Phi_{\mathcal{C}}^\top \Phi_{\mathcal{C}} + \alpha I\right)^{-1} \Phi_{\mathcal{C}}^\top \bar{q}^r$; $w^c \leftarrow \left(\Phi_{\mathcal{C}}^\top \Phi_{\mathcal{C}} + \alpha I\right)^{-1} \Phi_{\mathcal{C}}^\top \bar{q}^c$

8: $Q^r(s,a) \leftarrow \langle w^r, \phi(s,a) \rangle$, $Q^c(s,a) \leftarrow \langle w^c, \phi(s,a) \rangle$ for all $s, a$
    **return** $Q^r, Q^c$

---

Given a core set $\mathcal{C}$, a set of trajectories, a behaviour policy $\mu$, a target policy $\pi$, the LSE subroutine (algorithm 4) returns a least-square estimate $Q$ of $q_\pi$.

If the core set $\mathcal{C}$ is empty, we define $Q(\cdot, \cdot)$ to be zero. Then, for a target accuracy $\epsilon > 0$ and a uniform misspecification error $\omega$ defined in assumption 1, we have a bound on the accuracy of $\bar{q}$ with respect to $q_\pi$ as given by the next lemma.

**Lemma 5** *[Lemma 4.3 of Weisz et al. (2022)] Let $\pi$ be a randomized policy. Let $\mathcal{C} = \{(s_i, a_i)\}_{i \in [N]}$ be a set of state-action pairs of set size $N \in \mathbb{N}$. Assume for all $i \in [N]$, $|\bar{q}(s_i, a_i) - q_\pi(s_i, a_i)| \leq \epsilon$. Then, for all $s, a \in \mathcal{S} \times \mathcal{A}$,*

$$|Q(s,a) - q_\pi(s,a)| \leq \omega + \|\phi(s,a)\|_{V(\mathcal{C},\alpha)^{-1}} \left( \sqrt{\alpha}B + (\omega + \epsilon)\sqrt{N} \right).$$

Proof: If assumption 1 holds, then there exists a $w_\pi \in \mathbb{R}^d$, $\|w_\pi\|_2 \leq B$ such that for any $(s,a) \in \mathcal{S} \times \mathcal{A}$, $|\phi(s,a)^\top w_\pi - q_\pi(s,a)| \leq \omega$, where $\omega$ is the misspecification error. Let $\bar{w}_\pi = V(\mathcal{C}, \alpha)^{-1} \sum_{i \in [N]} \phi(s_i, a_i)\phi(s_i, a_i)^\top w_\pi$. For any $(s,a) \in \mathcal{S} \times \mathcal{A}$, recall $Q(s,a) = \phi(s,a)^\top w$, where $w = V(\mathcal{C}, \alpha)^{-1} \sum_{i \in [N]} \phi(s_i, a_i)\bar{q}(s_i, a_i)$. It follows that

$$|Q(s,a) - q_\pi(s,a)|$$
$$\leq |\phi(s,a)^\top(w - \bar{w}_\pi)| + |\phi(s,a)^\top(\bar{w}_\pi - w_\pi)| + |\phi(s,a)^\top w_\pi - q_\pi(s,a)|. \tag{14}$$

To bound the second term in eq. (14), we have

$$\begin{aligned}
|\phi(s,a)^\top(\bar{w}_\pi - w_\pi)| &\leq \|\phi(s,a)\|_{V(\mathcal{C},\alpha)^{-1}} \|\bar{w}_\pi - w_\pi\|_{V(\mathcal{C},\alpha)} \\
&\leq \|\phi(s,a)\|_{V(\mathcal{C},\alpha)^{-1}} \| - \alpha V(\mathcal{C},\alpha)^{-1} w_\pi\|_{V(\mathcal{C},\alpha)} \\
&= \alpha\|\phi(s,a)\|_{V(\mathcal{C},\alpha)^{-1}} \|w_\pi\|_{V(\mathcal{C},\alpha)^{-1}} \\
&\leq \alpha\|\phi(s,a)\|_{V(\mathcal{C},\alpha)^{-1}} \|w_\pi\|_{\frac{1}{\alpha}I} \\
&\leq \alpha\|\phi(s,a)\|_{V(\mathcal{C},\alpha)^{-1}} \sqrt{\frac{1}{\alpha}}B = \sqrt{\alpha}B\|\phi(s,a)\|_{V(\mathcal{C},\alpha)^{-1}}.
\end{aligned} \tag{15}$$

Let $\alpha$ be the smallest eigenvalue of $V(\mathcal{C}, \alpha)$, then by eigendecomposition, $V(\mathcal{C}, \alpha) = Q\Lambda Q^\top \geq Q(\alpha I)Q^\top \geq \alpha QQ^\top \geq \alpha I$ since $QQ^\top$ is orthonormal. This implies that $V(\mathcal{C}, \alpha)^{-1} \leq \frac{1}{\alpha}I$, which leads to eq. (15).

Finally, we bound the first term in eq. (14). For every $i \in [N]$, let $\xi_i = \phi(s_i, a_i)^\top w_\pi - \bar{q}(s_i, a_i)$. Then,

$$|\xi_i| = |\bar{q}(s_i, a_i) - \phi(s_i, a_i)^\top w_\pi| \leq |\bar{q}(s_i, a_i) - q_\pi(s_i, a_i)| + |q_\pi(s_i, a_i) - \phi(s_i, a_i)^\top w_\pi|$$
$$\leq \epsilon + \omega.$$

It follows that for all $s, a \in \mathcal{S} \times \mathcal{A}$,

$$|\phi(s,a)^\top(w - \bar{w}_\pi)| = |\langle V(\mathcal{C},\alpha)^{-1} \sum_{i \in [N]} \phi(s_i, a_i) \left( \bar{q}(s_i, a_i) - \phi(s_i, a_i)^\top w_\pi \right), \phi(s,a) \rangle|$$

$$= |\langle V(\mathcal{C},\alpha)^{-1} \sum_{i \in [N]} \phi(s_i, a_i)\xi_i, \phi(s,a) \rangle|$$

$$\leq \sum_{i \in [N]} |\langle V(\mathcal{C},\alpha)^{-1}\phi(s_i, a_i)\xi_i, \phi(s,a) \rangle|$$

$$\leq (\omega + \epsilon) \sum_{i \in [N]} |\langle V(\mathcal{C},\alpha)^{-1}\phi(s_i, a_i), \phi(s,a) \rangle|$$

$$\leq (\omega + \epsilon)\sqrt{|\mathcal{C}|} \sqrt{\sum_{i \in [N]} \langle V(\mathcal{C},\alpha)^{-1}\phi(s_i, a_i), \phi(s,a) \rangle^2} \quad \text{by Holder's inequality}$$

$$\leq (\omega + \epsilon)\sqrt{|\mathcal{C}|} \sqrt{\phi(s,a)^\top V(\mathcal{C},\alpha)^{-1} \left( \sum_{i \in [N]} \phi(s_i, a_i)\phi(s_i, a_i)^\top \right) V(\mathcal{C},\alpha)^{-1}\phi(s,a)}$$

$$\leq (\omega + \epsilon)\sqrt{|\mathcal{C}|} \sqrt{\phi(s,a)^\top V(\mathcal{C},\alpha)^{-1} \left( \sum_{i \in [N]} \phi(s_i, a_i)\phi(s_i, a_i)^\top + \alpha I \right) V(\mathcal{C},\alpha)^{-1}\phi(s,a)}$$

$$= (\omega + \epsilon)\sqrt{|\mathcal{C}|} \sqrt{\phi(s,a)^\top V(\mathcal{C},\alpha)^{-1}\phi(s,a)}$$

$$= (\omega + \epsilon)\sqrt{N} \|\phi(s,a)\|_{V(\mathcal{C},\alpha)^{-1}}.$$

Putting everything together complete the proof. ∎

### A.3 The accuracy of least-square estimates

Given a core set $\mathcal{C}$ and a target policy $\pi$, for any $s \in \mathrm{Cov}(\mathcal{C}), a \in \mathcal{A}$, the feature vector $\phi(s, a)$ satisfies $\|\phi(s, a)\|_{V(\mathcal{C}, \alpha)^{-1}} \leq 1$. Then, by lemma 5, we have $|Q(s, a) - q_\pi(s, a)| = O(\omega + \epsilon)$ for any $s \in \mathrm{Cov}(\mathcal{C})$. In this section, we verify whether this accuracy is maintained throughout the execution of our algorithm.

We note that policy improvements can only occur during a running phase $\ell = l$. When all $(s, a)$ pairs in $\mathcal{C}_\ell$ have their placeholder value $\perp$ replaced by trajectories, algorithm 2 executes line 17 to line 27. During each iteration from $k_\ell$ to $k_{\ell+1} - 1$, the LSE subroutine is executed. The accuracy of $\bar{q}_k$ is used to bound the estimation error in lemma 5. Therefore, we will first verify that the accuracy guarantee of $\bar{q}_k(s, a)$ used in lemma 4 is indeed satisfied by the main algorithm and maintained throughout its execution.

Once a state-action pair is added to a core set, it remains in that core set for the duration of the algorithm. This means that any core set $\mathcal{C}_l$ for $l \in \{0, \dots, L + 1\}$ can grow in size over time. When a core set $\mathcal{C}_l$ is extended during a running phase $\ell = l - 1$, the least-square estimate will need be updated based on the newly extended $\mathcal{C}_l$ in running phase $\ell = l$, which contains newly discovered features. However, the policy is update only for states that are newly covered by the extended core set $\mathcal{C}_l$ using the newly improved estimates. Meanwhile, the policy for other states that have already been updated by a prior softmax update remain unchanged. Note that after line 27 of algorithm 2 is run, the next phase's core set $\mathcal{C}_{l+1}$ will be set to $\mathcal{C}_l$, which means that any state that was once newly covered by $\mathcal{C}_l$ is no longer considered newly covered. Consequently, the policy for those states will remain unchanged throughout the rest of the algorithm's execution. By updating the policies accordingly, we arrive at the following lemma, which will be crucial in proving the accuracy guarantee of the least-squares estimators.

**Lemma 6** *For any $l \in \{0, \dots, L\}$, let $\mathcal{C}_l^{past}$ be any past version of $\mathcal{C}_l$ and let $\pi_k^{past}$ for $k = k_l, \cdots, k_{l+1} - 1$ be the corresponding policies associated with $\mathcal{C}_l^{past}$. If at any later point during the execution of the algorithm, $\pi_k$ is updated again, then it holds that*

$$\pi_k \in \Pi_{\pi_k, \mathrm{Cov}(\mathcal{C}_l)} \subseteq \Pi_{\pi_k^{past}, \mathrm{Cov}(\mathcal{C}_l^{past})}.$$

*Additionally, for any states that have been covered by $\mathcal{C}_l^{past}$, it will continue to be covered by $\mathcal{C}_l$ throughout the execution of the algorithm. In other words, $s \in \mathrm{Cov}(\mathcal{C}_l^{past}) \subseteq \mathrm{Cov}(\mathcal{C}_l)$.*

Proof: The proof follows similar logic to Lemma 4.5 of Weisz et al. (2022). Recall for matrices $A, B$, $A \geq B$ means that $A - B$ is positive semidefinite.

Because $\mathcal{C}_l \supseteq \mathcal{C}_l^{past}$, there may be more rows added to $\Phi_{\mathcal{C}_l}$ than $\Phi_{\mathcal{C}_l^{past}}$. Recall $V(\mathcal{C}_l, \alpha) = \Phi_{\mathcal{C}_l}^\top \Phi_{\mathcal{C}_l} + \alpha I$, and likewise for $V(\mathcal{C}_l^{past})$ except $\Phi_{\mathcal{C}_l}$ is replaced by $\mathcal{C}_l^{past}$. Note that both $V(\mathcal{C}_l, \alpha), V(\mathcal{C}_l^{past}, \alpha)$ are dimension $d \times d$. Let $\mathcal{C}_l'$ contain a set of state-action pairs that are in $\mathcal{C}_l \setminus \mathcal{C}_l^{past}$. Then, $V(\mathcal{C}_l, \alpha) - V(\mathcal{C}_l^{past}, \alpha) = \sum_{(s,a) \in \mathcal{C}_l'} \phi(s, a)\phi(s, a)^\top$. Since any rank-1 matrices is positive semidefinite and their sum is also positive semidefinite, it follows that $V(\mathcal{C}_l, \alpha) \geq V(\mathcal{C}_l^{past}, \alpha)$.

Since $V(\mathcal{C}_l, \alpha)$ and $V(\mathcal{C}^{past}, \alpha)$ are symmetric positive definite matrices, then it follows that $V(\mathcal{C}_l, \alpha)^{-1} \leq V(\mathcal{C}_l^{past}, \alpha)^{-1}$. From this, we see that for any $x \in \mathbb{R}^d$, $\|x\|_{V(\mathcal{C}_l, \alpha)^{-1}} \leq \|x\|_{V(\mathcal{C}_l^{past}, \alpha)^{-1}}$. Then, it follows that $\mathrm{ActionCov}(\mathcal{C}_l^{past}) \subseteq \mathrm{ActionCov}(\mathcal{C}_l)$, and likewise $\mathrm{Cov}(\mathcal{C}_l^{past}) \subseteq \mathrm{Cov}(\mathcal{C}_l)$. Therefore, for an $s \in \mathrm{Cov}(\mathcal{C}_l^{past})$, the same state $s \in \mathrm{Cov}(\mathcal{C}_l)$, and the second result follows. Finally, by the definition of the extended policy set definition 1, $\Pi_{\pi_k, \mathrm{Cov}(\mathcal{C}_l)} \subseteq \Pi_{\pi_k^{past}, \mathrm{Cov}(\mathcal{C}_l^{past})}$. ∎

**Lemma 7** *For any $l \in \{0, \dots, L\}$ and any $(s, a) \in \mathcal{C}_l$, the importance-weighted $\bar{q}_k(s, a)$ computed in the LSE subroutine during the running phase $\ell = l$ is an unbiased estimator of the expected discounted reward:*

$$E_{\pi_k', s, a}\left[\sum_{h=0}^{H-1} \gamma^h R_{h+1}\right] \quad \text{for } \pi_k' \in \Pi_{\pi_k, \mathrm{Cov}(\mathcal{C}_l)},$$

*for iterations $k = k_l, \cdots, k_{l+1} - 1$ associated with this phase.*

Proof: For the algorithm to execute the LSE subroutine, every $(s, a) \in \mathcal{C}_l$ must have its placeholder value $\bot$ in $D_l[(s, a)]$ replaced with trajectories. Trajectories are stored in $D_l$ only when the Gather-data subroutine returns "discovered is False" during the running phase $\ell = l$. This ensures that every state within these trajectories has passed the uncertainty test, thereby ensuring that all such states are in the cover of $\mathcal{C}_l$ and will remain in the cover of $\mathcal{C}_l$ for the duration of the algorithm, as established in lemma 6. Additionally, once trajectories for a $(s, a) \in \mathcal{C}_l$ are stored in $D_l$, they remain unchanged throughout the algorithm's execution. We aim to show that $\bar{q}_k(s, a)$ computed using $D_l[(s, a)]$, is an unbiased estimate of the stated quantity for all iterations associated with the phase. This will be established through the following inductive arguments.

**Base case:** for a $(s, a) \in \mathcal{C}_l$, the trajectories are generated and stored in $D_l[(s, a)]$ for the first time during the running phase $\ell = l$.

We let $\tau_{s,a}^i$ denote the $i$-th trajectory $(S_0^i = s, A_0^i = a, R_1^i, S_1^i, \ldots, S_{H-1}^i, A_{H-1}^i, R_H^i)$ generated by $\pi_{k_l}$ interacting with the simulator, and there are $n$ such trajectories stored in $D_l[(s, a)]$. Then, for all $k = k_l, \cdots, k_{l+1} - 1$, the return

$$\bar{q}_k(s, a) = \frac{1}{n} \sum_{i=1}^n \Pi_{h=1}^{H-1} \frac{\pi_k(A_h^i | S_h^i)}{\pi_{k_l}(A_h^i | S_h^i)} G(\tau_{s,a}^i),$$

where $G(\tau_{s,a}^i) = \sum_{h=0}^{H-1} \gamma^h R_{h+1}$.

The behavior policy $\pi_{k_l}$ is updated in a previous loop through the algorithm when $\ell = l - 1$. For iterations, starting with $k = k_l + 1, \ldots, k_{l+1} - 1$, the policy $\pi_k$ is updated in iteration $k - 1$. Thus, the most recent policy $\pi_k$ and the behaviour policy $\pi_{k_l}$ are available for the computation of the importance sampling ratio: $\rho_k(\tau_{s,a}^i) = \Pi_{h=1}^{H-1} \frac{\pi_k(A_h^i | S_h^i)}{\pi_{k_l}(A_h^i | S_h^i)}$. We show that the importance weighted return $\rho_k(\tau_{s,a}^i) G(\tau_{s,a}^i)$ is an unbiased estimate of $\mathbb{E}_{\pi_k, s, a}[G(\tau_{s,a}^i)]$:

$$\mathbb{E}_{\pi_{k_l}, s, a} \left[ \rho_k(\tau_{s,a}^i) \sum_{h=0}^{H-1} \gamma^h R_{h+1}^i \right]$$

$$= \mathbb{E}_{\pi_{k_l}, s, a} \left[ \frac{\delta(s, a) P(S_1 | S_0 = s, A_0 = a) \pi_k(A_1 | S_1) \ldots . \pi_k(A_{H-1} | S_{H-1})}{\delta(s, a) P(S_1 | S_0 = s, A_0 = a) \pi_{k_l}(A_1 | S_1) \ldots \pi_{k_l}(A_{H-1} | S_{H-1})} \sum_{h=0}^{H-1} \gamma^h R_{h+1}^i \right]$$

$$= \mathbb{E}_{\pi_k, s, a} \left[ \sum_{h=0}^{H-1} \gamma^h R_{h+1}^i \right],$$

where $\delta(s, a)$ is the dirac-delta function. Note, for the on-policy iteration $k = k_l$, the importance sampling ratio $\rho_k(\tau_{s,a}^i) = 1$, and the result is trivially satisfied.

Finally, since all the states in the trajectories are in $\text{Cov}(\mathcal{C}_l)$, it follows that any policy $\pi_k' \in \Pi_{\pi_k, \text{Cov}(\mathcal{C}_l)}$ produces the same $\rho_k(\tau_{s,a}^i)$. The return $\rho_k(\tau_{s,a}^i) G(\tau_{s,a}^i)$ is an unbiased estimate of $E_{\pi_k', s, a}[G(\tau_{s,a}^i)]$ for all $\pi_k' \in \Pi_{\pi_k, \text{Cov}(\mathcal{C}_l)}$. This is true for all $i = 1, \ldots, n$. Consequently, $\bar{q}_k(s, a)$ is an unbiased estimate of $\mathbb{E}_{\pi_k', s, a}[\sum_{h=0}^{H-1} R_{h+1}]$ for all for all $\pi_k' \in \Pi_{\pi_k, \text{Cov}(\mathcal{C}_l)}$.

The requirement that the importance weighted $\bar{q}_k$ be unbiased for all policies $\pi_k' \in \Pi_{\pi_k, \text{Cov}(\mathcal{C}_l)}$ is important. This ensures that if $\pi_k$ is to be updated in a future loop through the algorithm again, the estimates remain unbiased and unchanged.

**Previously generated trajectories:** for any $(s, a) \in \mathcal{C}_l$, the trajectories have already been generated and stored in $D_l[(s, a)]$ during a previous loop of the algorithm when $\ell = l$.

Let $D_l^{\text{past}}[(s, a)]$ denote a past snapshot of the data stored for a $(s, a) \in \mathcal{C}_l^{\text{past}}$. Let $\pi_k^{\text{past}}$ for $k = k_l, \ldots k_{l+1} - 1$ denote the policies associated with $\mathcal{C}_l^{\text{past}}$ after line 27 has been run. Finally, let $\tau_{s,a}^{i;\text{past}}$ denote the $i$-th trajectory stored in $D_l^{\text{past}}[(s, a)]$.

Assume the importance weighted return $\rho_k(\tau_{s,a}^{i,\text{past}}) G(\tau_{s,a}^{i,\text{past}})$ is an unbiased estimate of $\mathbb{E}_{\tilde{\pi}_k, s, a}[G(\tau_{s,a}^{i,\text{past}})]$ for all $\tilde{\pi}_k \in \Pi_{\pi_k^{\text{past}}, \text{Cov}(\mathcal{C}_l^{\text{past}})}$ for $k = k_l, \ldots, k_{l+1} - 1$. When the algorithm executes a loop with $\ell = l$ again, by lemma 6, the most recent policy $\pi_k \in \Pi_{\pi_k, \text{Cov}(\mathcal{C}_l)} \subseteq \Pi_{\pi_k^{\text{past}}, \text{Cov}(\mathcal{C}_l^{\text{past}})}$, then $\rho_k(\tau_{s,a}^{i,\text{past}}) G(\tau_{s,a}^{i,\text{past}})$ is also an unbiased estimate of $\mathbb{E}_{\pi_k', s, a}[G(\tau_{s,a}^{i,\text{past}})]$ for all $\pi_k' \in \Pi_{\pi_k, \text{Cov}(\mathcal{C}_l)}$.

Once $D_l^{\text{past}}[(s,a)]$ is populated with trajectories, $D_l^{\text{past}}[(s,a)]$ remain unchanged throughout the execution of the algorithm. Therefore, $G(\tau_{s,a}^i) = G(\tau_{s,a}^{i,\text{past}})$. Since all the states in the trajectories are in $\text{Cov}(\mathcal{C}_l^{\text{past}}) \subseteq \text{Cov}(\mathcal{C}_l)$ by lemma 6, any policy $\tilde{\pi}_k \in \Pi_{\pi_k^{\text{past}}, \text{Cov}(\mathcal{C}_l^{\text{past}})}$ produces the same $\rho_k(\tau_{s,a}^i)$. Thus, we have $\rho_k(\tau_{s,a}^i)G(\tau_{s,a}^i) = \rho_k(\tau_{s,a}^{i,\text{past}})G(\tau_{s,a}^{i,\text{past}})$. It follows that $\rho_k(\tau_{s,a}^i)G(\tau_{s,a}^i)$ is an unbiased estimate of $\mathbb{E}_{\pi_k',s,a}[G(\tau_{s,a}^i)]$ for all $\pi_k' \in \Pi_{\pi_k, \text{Cov}(\mathcal{C}_l)}$. This is true for all $i = 1, \ldots, n$. Consequently, $\bar{q}_k(s,a)$ is an unbiased estimate of $\mathbb{E}_{\pi_k',s,a}[\sum_{h=0}^{H-1} R_{h+1}]$ for all $\pi_k' \in \Pi_{\pi_k, \text{Cov}(\mathcal{C}_l)}$. $\blacksquare$

**Lemma 8** *Whenever the LSE-subroutine of Confident-NPG is executed during a running phase $\ell = l$ for $l \in \{0, \ldots, L\}$, the behaviour policy $\pi_{k_\ell}(\cdot|s)$ and target policy $\pi_k(\cdot|s)$ for $k = k_\ell, \ldots, k_{\ell+1} - 1$ satisfy eq. (9) for any $s \in \text{Cov}(\mathcal{C}_\ell)$.*

Proof: Recall that the behavior policy $\pi_{k_l}$ is updated during a previous loop of the algorithm when $\ell = l - 1$. By the time the LSE subroutine is executed, $\pi_{k_\ell}$ will be the policy that generated the data. Therefore, for the on-policy iteration where $k = k_\ell$, eq. (9) is trivially satisfied.

For subsequent iterations, starting with $k = k_\ell + 1, \ldots, k_{\ell+1} - 1$, for any $s \in \text{Cov}(\mathcal{C}_\ell)$, the policy $\pi_k(\cdot|s)$ will have either performed a softmax update for the first time or remain unchanged from a previous softmax update based on an earlier least-square estimate. Either way, for any $s \in \text{Cov}(\mathcal{C}_\ell)$, the target policy $\pi_k$ and behaviour policy $\pi_{k_\ell}$ relate to each other in the form of $\pi_k(\cdot|s) \propto \pi_{k_\ell}(\cdot|s) \exp(\eta_1 \sum_{t=k_\ell}^{k-1} \tilde{Q}_t(s,a))$. Since $\tilde{Q}_t(s,a) \in [0, \frac{1}{1-\gamma}]$ for any $t = k_\ell, \ldots, k-1$, then it follows that

$$0 \leq \eta_1 \sum_{t=k_\ell}^{k-1} \tilde{Q}_t(s,a) \leq \eta_1(k - k_\ell)\frac{1}{1-\gamma} \leq \frac{\eta_1((\lfloor m \rfloor + 1) - 1)}{1-\gamma}.$$

By choosing $\eta_1 = (1-\gamma)\sqrt{\frac{2\ln(|\mathcal{A}|)}{K}}, H = \frac{\ln(4/\epsilon(1-\gamma))}{1-\gamma}, m = \frac{\ln(1+\rho_0)}{2H\epsilon(1-\gamma)^2}$, and $K = \frac{2\ln(A)}{(1-\gamma)^4\epsilon^2}$, we have $\frac{\eta_1((\lfloor m \rfloor + 1) - 1)}{1-\gamma} \leq \frac{\eta_1 m}{1-\gamma} = \frac{\ln(1+\rho_0)}{2H}$. Then it follows that eq. (9) is satisfied. $\blacksquare$

**Lemma 9** *For any $l \in \{0, \ldots, L\}$ and any $(s,a) \in \mathcal{C}_l$, the importance-weighted $\bar{q}_k(s,a)$ computed in the LSE subroutine during the running phase $\ell = l$ satisfies the following with probability $1 - \delta'$,*

$$|\bar{q}_k(s,a) - q_{\pi_k'}(s,a)| \leq \epsilon \quad \text{for } \pi_k' \in \Pi_{\pi_k, \text{Cov}(\mathcal{C}_l)} \tag{16}$$

*for all iterations $k = k_l, \ldots, k_{l+1} - 1$ associated with this phase.*

Proof: We apply lemma 4 to each $(s,a) \in \mathcal{C}_l$. To ensure the applicability of the lemma, we verify its two conditions: 1) the policies satisfy eq. (9) and 2) the estimate are unbiased.

We note that when the LSE-subroutine is executed during a running phase with $\ell = l$, the Gather-data subroutine has already completed, and the algorithm trajectories for each state-action pair $(s,a) \in \mathcal{C}_l$ are stored in $D_l[(s,a)]$. For any trajectory to be stored in $D_l$, this means that every state within the trajectories has passed the uncertainty test, ensuring that all such states are in the cover of $\mathcal{C}_l$. By lemma 6, these states will continue to be covered by $\mathcal{C}_l$ throughout the execution of the algorithm. The implication of this is that all the states in a trajectory of $D_l[(s,a)]$ satisfy eq. (9) by lemma 8.

Second, by lemma 7, the importance weighted return $\bar{q}_k(s,a)$ is unbiased estimate of any $\mathbb{E}_{\pi_k',s,a}\left[\sum_{h=0}^{H-1} \gamma^h R_{h+1}\right]$ for all $\pi_k' \in \Pi_{\pi_k, \text{Cov}(\mathcal{C}_l)}$. Altogether, by lemma 4, we can ensure eq. (16) holds.

Consider a past loop through the algorithm with $\ell = l$, let $\mathcal{C}_l^{\text{past}}$ be the core set and $\pi_k^{\text{past}}$ for $k = k_l, \ldots, k_{l+1} - 1$ be the policies associated with $\mathcal{C}_l^{\text{past}}$ after line 27 has been run. If eq. (16) holds for all $\tilde{\pi}_k \in \Pi_{\pi_k^{\text{past}}, \text{Cov}(\mathcal{C}_l^{\text{past}})}$, then the accuracy of $\bar{q}_k$ will continue to hold for any future update of $\pi_k$ because $\pi_k \in \Pi_{\pi_k, \text{Cov}(\mathcal{C}_l)} \subseteq \Pi_{\pi_k^{\text{past}}, \text{Cov}(\mathcal{C}_l^{\text{past}})}$ by lemma 6. $\blacksquare$

**Lemma 10 (Weisz et al. (2022))** *At any time during the execution of the main algorithm, for all $l \in \{0, \ldots, L\}$, the size of each $\mathcal{C}_l$ is bounded:*

$$|\mathcal{C}_l| \leq 4d \ln\left(1 + \frac{4}{\alpha}\right) = \tilde{d} = \tilde{O}(d),$$

*where the $\alpha$ is the smallest eigenvalue of $V(\mathcal{C}, \alpha)$ and $N$ is the radius of the Euclidean ball containing all the feature vectors.*

**Lemma 11** *Whenever LSE subroutine of Confident-NPG is executed during a running phase $\ell = l$ for $l \in \{0, \ldots, L\}$, the least-square estimate $\tilde{Q}_k(s, a)$ satisfies the following condition for all iterations $k = k_\ell, \cdots, k_{\ell+1} - 1$ associated with this phase and for all $s \in \text{Cov}(C_\ell)$ and $a \in \mathcal{A}$,*

$$|\tilde{Q}_k(s, a) - q_{\pi'_k}(s, a)| \leq \epsilon' \quad \text{for all } \pi'_k \in \Pi_{\pi_k, \text{Cov}(\mathcal{C}_\ell)}, \tag{17}$$

*where $\epsilon' = \omega + \sqrt{\alpha}B + (\omega + \epsilon)\sqrt{\tilde{d}}$.*

Proof: We prove the result by induction similar to Lemma F.1 of Weisz et al. (2022). We let $\mathcal{C}_l^-, \pi_k^-, \tilde{Q}_k^-$ to denote the value of variable $\mathcal{C}_l, \pi_k, \tilde{Q}_k$ at the time when line 17 to line 27 were most recently executed with $\ell = l$ in a previous loop through the algorithm. If such time does not exist, we let their values be the initialization values. Only after the execution of line line 27 will $\mathcal{C}_l^-$ change and as well as $\mathcal{C}_{l+1}$, and this is the only time that $\mathcal{C}_{l+1}$ can be changed. Therefore, at the start of a new loop, we see that $\mathcal{C}_{l+1} = \mathcal{C}_l^-$. This also holds at the initialization of the algorithm, we conclude that at the start of each loop, $\text{Cov}(\mathcal{C}_{l+1}) = \text{Cov}(\mathcal{C}_l^-)$.

At initialization, $\tilde{Q}_k = 0$ for any $k \in \{0, \ldots, K\}$ and $C_l = ()$ for all $l \in \{0, \ldots, L\}$. By applying lemma 5 (Lemma 4.3 of Weisz et al. (2022)), for any $(s, a) \in \mathcal{S} \times \mathcal{A}$,

$$|\tilde{Q}_k(s, a) - q_{\pi'_k}(s, a)| \leq \omega + \sqrt{\alpha}B \leq \epsilon',$$

which satisfies eq. (17).

Next, let us consider the start of a loop after $\ell = l$ is set and assume that the inductive hypothesis holds for the previous time line 17 to line 27 were executed with the same value of $\ell = l$. For any $s \in \text{Cov}(\mathcal{C}_{l-1}^-)$, policy $\pi_{k_l}(\cdot|s)$ would have already been set in a previous loop with value $l - 1$ and remains unchanged in the current loop. By lemma 9, the condition of lemma 5 holds, then by lemma 5, we have for any $s \in \text{Cov}(\mathcal{C}_{l-1}^-)$,

$$|\tilde{Q}_{k_l}^-(s, \cdot) - q_{\pi'_{k_l}}(s, \cdot)| \leq \omega + \sqrt{\alpha}B + (\omega + \epsilon)\sqrt{\tilde{d}} \quad \text{for } \pi_{k'_l} \in \Pi_{\pi_{k_l}^-, \text{Cov}(\mathcal{C}_{l-1}^-)},$$

where $\|\phi(s, \cdot)\|_{V(\mathcal{C}_{l-1}, \alpha)^{-1}} \leq 1$ because $s \in \text{Cov}(\mathcal{C}_{l-1}^-)$ and $|C_{l-1}^-| \leq \tilde{d}$ by lemma 10. Recall by definition, $\tilde{Q}_{k_l} = \tilde{Q}_{k_l}^-, \pi_{k_l} = \pi_{k_l}^-, C_l = C_{l-1}^-$, and $\text{Cov}(\mathcal{C}_l) = \text{Cov}(\mathcal{C}_{l-1}^-)$. It follows that for any $s \in \text{Cov}(\mathcal{C}_l), |\tilde{Q}_{k_l}(s, \cdot) - q_{\pi'_{k_l}}(s, \cdot)| \leq \epsilon'$ for $\pi_{k'_l} \in \Pi_{\pi_{k_l}, \text{Cov}(\mathcal{C}_l)}$.

For any $s$ that is already covered by $\mathcal{C}_l$ (i.e., $s \in \text{Cov}(\mathcal{C}_l^-)$), and for any off-policy iteration $k = k_l + 1, \cdots, k_{l+1} - 1$, $\tilde{Q}_k(s, \cdot) = \tilde{Q}_k^-(s, \cdot)$. Additionally, the policy $\pi_k(\cdot|s)$ would already have been set in a previous running loop with the same value of $l$ and remains unchanged in the current loop. For $s \in \text{Cov}(\mathcal{C}_l^-)$, by lemma 9, the condition of lemma 5 holds, and then by lemma 5,

$$|\tilde{Q}_k^-(s, \cdot) - q_{\pi'_k}(s, \cdot)| \leq \omega + \sqrt{\alpha}B + (\omega + \epsilon)\sqrt{\tilde{d}} \quad \text{for } \pi_{k'} \in \Pi_{\pi_k^-, \text{Cov}(\mathcal{C}_l^-)},$$

where $\|\phi(s, \cdot)\|_{V(\mathcal{C}_l^-, \alpha)^{-1}} \leq 1$ because $s \in \text{Cov}(\mathcal{C}_l^-)$ and $|\mathcal{C}_l^-| \leq \sqrt{\tilde{d}}$ by lemma 10. By lemma 6, $\Pi_{\pi_k, \text{Cov}(\mathcal{C}_l)} \subseteq \Pi_{\pi_k^-, \text{Cov}(\mathcal{C}_l^-)}$. By definition, $\tilde{Q}_k(s, \cdot) = \tilde{Q}_k^-(s, \cdot)$ for $s \in \text{Cov}(\mathcal{C}_{l+1}) = \text{Cov}(\mathcal{C}_l^-)$, $|\tilde{Q}_k(s, \cdot) - q_{\pi'_k}(s, \cdot)| \leq \epsilon'$ for any $\pi'_k \in \Pi_{\pi_k, \text{Cov}(C_{l+1})}$.

Finally, for any $s$ that is newly covered by $\mathcal{C}_l$ (i.e., $s \notin \text{Cov}(\mathcal{C}_{l+1})$), and for all $k = k_l, \ldots, k_{l+1} - 1$, $\tilde{Q}_k(s, \cdot) = Q_k(s, \cdot)$. By lemma 9, the condition of lemma 5 holds, and then by lemma 5, we have

$$|Q_k(s, \cdot) - q_{\pi'_k}(s, \cdot)| \leq \omega + \sqrt{\alpha}B + (\omega + \epsilon)\sqrt{\tilde{d}} \quad \text{for } \pi_{k'} \in \Pi_{\pi_k, \text{Cov}(\mathcal{C}_l)},$$

where $\|\phi(s, \cdot)\|_{V(\mathcal{C}_l, \alpha)^{-1}} \leq 1$ and $|\mathcal{C}_l| \leq \tilde{d}$ by lemma 10. ∎

**Lemma 12** *For any $\delta' \in (0, 1]$, a target accuracy $\epsilon > 0$, misspecification error $\omega \geq 0$, and initial state $s_0$, with probability at least $1 - \delta'$, the value difference between any $\pi \in \Pi_{rand}$ and the mixture policy $\bar{\pi}_K$ returned by Confident-NPG has the following bound:*

$$v_\pi(s_0) - v_{\bar{\pi}_K}(s_0) \leq \frac{4\epsilon'}{1 - \gamma} + \frac{1}{K(1 - \gamma)} \sum_{k=0}^{K-1} \mathbb{E}_{s' \sim d_\pi(s_0), s' \in \text{Cov}(\mathcal{C}_0)} \left[ \langle \tilde{Q}_k(s', \cdot), \pi(\cdot|s') - \pi_k(\cdot|s') \rangle \right].$$

Proof: For any $l \in \{0, \dots, L\}$ and for all iterations $k = k_l, \dots, k_{l+1} - 1$ associated with $l$, define

$$\pi_k^+(\cdot|s) = \begin{cases} \pi_k(\cdot|s) & \text{if } s \in \text{Cov}(\mathcal{C}_l) \\ \pi(\cdot|s) & \text{otherwise.} \end{cases}$$

Then, for any $s \in \text{Cov}(\mathcal{C}_l)$,

$$v_\pi(s) - v_{\pi_k}(s) = v_\pi(s) - v_{\pi_k^+}(s) + v_{\pi_k^+}(s) - v_{\pi_k}(s)$$

$$= \underbrace{\frac{1}{1-\gamma} \mathbb{E}_{s' \sim d_\pi(s)} \left[ \langle q_{\pi_k^+}(s', \cdot), \pi(\cdot|s') - \pi_k^+(\cdot|s') \rangle \right]}_{I} \quad \text{by performance difference lemma}$$

$$+ \underbrace{\langle q_{\pi_k^+}(s, \cdot), \pi_k^+(\cdot|s) \rangle - \langle q_{\pi_k}(s, \cdot), \pi_k(\cdot|s) \rangle}_{II},$$

where $d_\pi(s)$ is the discounted state occupancy measure induced by following $\pi$ starting from $s$.

To bound term $II$, we note that for any $s \in \text{Cov}(\mathcal{C}_l)$, we have $\pi_k^+(\cdot|s) = \pi_k(\cdot|s)$ and both $\pi_k, \pi_k^+(\cdot|s) \in \Pi_{\pi_k, \text{Cov}(\mathcal{C}_l)}$. By lemma 11, we have for any $s \in \text{Cov}(\mathcal{C}_l), a \in \mathcal{A}, |\tilde{Q}_k(s, a) - q_{\pi_k'}(s, a)| \le \epsilon'$ for any $\pi_k' \in \Pi_{\pi_k, \text{Cov}(\mathcal{C}_l)}$. Then, for any $s \in \text{Cov}(\mathcal{C}_l), a \in \mathcal{A}$,

$$|q_{\pi_k^+}(s, a) - q_{\pi_k}(s, a)| \le |q_{\pi_k^+}(s, a) - \tilde{Q}_k(s, a)| + |\tilde{Q}_k(s, a) - q_{\pi_k}(s, a)| \le 2\epsilon'.$$

It follows that for any $s \in \text{Cov}(\mathcal{C}_l)$,

$$\langle q_{\pi_k^+}(s, \cdot), \pi_k^+(\cdot|s) \rangle - \langle q_{\pi_k}(s, \cdot), \pi_k(\cdot|s) \rangle = \langle \pi_k(\cdot|s), q_{\pi_k^+}(s, \cdot) - q_{\pi_k}(s, \cdot) \rangle$$

$$\le |\langle \pi_k(\cdot|s), q_{\pi_k^+}(s, \cdot) - q_{\pi_k}(s, \cdot) \rangle|$$

$$\le \|q_{\pi_k^+}(s, \cdot) - q_{\pi_k}(s, \cdot)\|_\infty \|\pi_k(\cdot|s)\|_1$$

$$\le 2\epsilon'.$$

To bound term $I$, we note that for any $s \notin \text{Cov}(\mathcal{C}_l), \pi_k^+(\cdot|s) = \pi(\cdot|s)$ and $\pi_k^+ \in \Pi_{\pi_k, \text{Cov}(\mathcal{C}_l)}$, then

$$\frac{1}{1-\gamma} \mathbb{E}_{s' \sim d_s^\pi} \left[ \langle q_{\pi_k^+}(s', \cdot), \pi(\cdot|s') - \pi_k^+(\cdot|s') \rangle \right]$$

$$= \frac{1}{1-\gamma} \mathbb{E}_{s' \sim d_s^\pi, s' \in \text{Cov}(\mathcal{C}_l)} \left[ \langle q_{\pi_k^+}(s', \cdot), \pi(\cdot|s') - \pi_k^+(\cdot|s') \rangle \right]$$

$$+ \frac{1}{1-\gamma} \mathbb{E}_{s' \sim d_s^\pi, s' \notin \text{Cov}(\mathcal{C}_l)} \left[ \langle q_{\pi_k^+}(s', \cdot), \pi(\cdot|s') - \pi_k^+(\cdot|s') \rangle \right]$$

$$= \frac{1}{1-\gamma} \mathbb{E}_{s' \sim d_s^\pi, s' \in \text{Cov}(\mathcal{C}_l)} \left[ \langle q_{\pi_k^+}(s', \cdot), \pi(\cdot|s') - \pi_k^+(\cdot|s') \rangle \right]$$

$$= \frac{1}{1-\gamma} \mathbb{E}_{s' \sim d_s^\pi, s' \in \text{Cov}(\mathcal{C}_l)} \left[ \langle q_{\pi_k^+}(s', \cdot) - \tilde{Q}_k(s', \cdot), \pi(\cdot|s') - \pi_k^+(\cdot|s') \rangle \right]$$

$$+ \frac{1}{1-\gamma} \mathbb{E}_{s' \sim d_s^\pi, s' \in \text{Cov}(\mathcal{C}_l)} \left[ \langle \tilde{Q}_k(s', \cdot), \pi(\cdot|s') - \pi_k^+(\cdot|s') \rangle \right]$$

$$\le \frac{1}{1-\gamma} \mathbb{E}_{s' \sim d_s^\pi, s' \in \text{Cov}(\mathcal{C}_l)} \left[ \|q_{\pi_k^+}(s', \cdot) - \tilde{Q}_k(s', \cdot)\|_\infty \|\pi(\cdot|s') - \pi_k^+(\cdot|s')\|_1 \right] \quad \text{by Holder's inequality}$$

$$+ \frac{1}{1-\gamma} \mathbb{E}_{s' \sim d_s^\pi, s' \in \text{Cov}(\mathcal{C}_l)} \left[ \langle \tilde{Q}_k(s', \cdot), \pi(\cdot|s') - \pi_k^+(\cdot|s') \rangle \right]$$

$$\le \frac{2\epsilon'}{1-\gamma} \quad \text{by lemma 11 and } \|\pi^*(\cdot|s') - \pi_k^+(\cdot|s')\|_1 \le 2$$

$$+ \frac{1}{1-\gamma} \mathbb{E}_{s' \sim d_s^\pi, s' \in \text{Cov}(\mathcal{C}_l)} \left[ \langle \tilde{Q}_k(s', \cdot), \pi(\cdot|s') - \pi_k(\cdot|s') \rangle \right]$$

$$= \frac{2\epsilon'}{1-\gamma} + \frac{1}{1-\gamma} \mathbb{E}_{s' \sim d_s^\pi, s' \in \text{Cov}(\mathcal{C}_l)} \left[ \langle \tilde{Q}_k(s', \cdot), \pi(\cdot|s') - \pi_k(\cdot|s') \rangle \right]$$

In summary, for any $l$, for any $k = k_l, \ldots, k_{l+1} - 1$ associated with $l$, and for any $s \in \text{Cov}(\mathcal{C}_l)$,

$$v_\pi(s) - v_{\pi_k}(s) \le \frac{4\epsilon'}{1 - \gamma} + \frac{1}{1 - \gamma} \mathbb{E}_{s' \sim d_{\pi(s)}, s' \in \text{Cov}(\mathcal{C}_l)} \left[ \langle \tilde{Q}_k(s', \cdot), \pi(\cdot|s') - \pi_k(\cdot|s') \rangle \right].$$

Because of line 27 of algorithm 2, one can use induction to show that by the time Confident-NPG terminates, all the $\mathcal{C}_l$ for $l \in \{0, \ldots, L+1\}$ will be equal. Therefore, the cover of $\mathcal{C}_l$ for all $l \in \{0, \ldots, L+1\}$ are also equal. Thus, it is sufficient to only consider $\mathcal{C}_0$ at the end of the algorithm. Because of line 3 of algorithm 2, $s_0 \in \text{Cov}(\mathcal{C}_0)$. Putting everything together, the value difference can be bounded as follows,

$$\frac{1}{K} \sum_{k=0}^{K-1} (v_\pi(s_0) - v_{\pi_k}(s_0)) = \frac{1}{K} \sum_{l=0}^{L} \sum_{k=k_l}^{k_{l+1}-1} (v_\pi(s_0) - v_{\pi_k}(s_0))$$

$$\le \frac{1}{K} \sum_{l=0}^{L} \sum_{k=k_l}^{k_{l+1}-1} \frac{4\epsilon'}{1-\gamma}$$

$$+ \frac{1}{K(1-\gamma)} \sum_{l=0}^{L} \sum_{k=k_l}^{k_{l+1}-1} \mathbb{E}_{s' \sim d_\pi(s_0), s' \in \text{Cov}(\mathcal{C}_l)} \left[ \langle \tilde{Q}_k(s', \cdot), \pi(\cdot|s') - \pi_k(\cdot|s') \rangle \right]$$

$$\le \frac{4\epsilon'}{1-\gamma} + \frac{1}{K(1-\gamma)} \sum_{k=0}^{K-1} \mathbb{E}_{s' \sim d_\pi(s_0), s' \in \text{Cov}(\mathcal{C}_0)} \left[ \langle \tilde{Q}_k(s', \cdot), \pi(\cdot|s') - \pi_k(\cdot|s') \rangle \right].$$

∎

## B  Confident-NPG-CMDP

We include the proofs of lemmas that appear in prior works and supporting lemmas that are helpful proving the lemmas in the main section. The lemmas that appear in the main section will have the same numbering here.

### B.1  The accuracy of least-square estimates

Once a state-action pair is added to a core set, it remains in that core set for the duration of the algorithm. This means that any $\mathcal{C}_l$ for $l \in \{0, \ldots, L+1\}$ can grow in size. When a core set $\mathcal{C}_l$ is extended during a running phase $\ell = l$, the least-square estimates will need be updated based on the newly extended $\mathcal{C}_l$ which contains newly discovered features. However, the policy is update only for states that are newly covered by the extended core set $\mathcal{C}_l$ using the newly improved estimates. Note that after line 30 of algorithm 1 is run, the next phase's core set $\mathcal{C}_{l+1}$ will be set to $\mathcal{C}_l$, which means that any state that was once newly covered by $\mathcal{C}_l$ is no longer considered newly covered. Consequently, the policy for those states will remain unchanged throughout the rest of the algorithm's execution.

We introduce hypothetical $\tilde{Q}_k^r$ and $\tilde{Q}_k^c$ to reflect the value of $\tilde{Q}_k^p$, used in the update of $\pi_{k+1}$ for $k = k_l, \ldots, k_{l+1} - 1$ associated with running phase $\ell = l$. At initialization, $\tilde{Q}_k^r(s,a) = 0, \tilde{Q}_k^c(s,a) = 0$ for all $k = 0, \ldots, K, s \in \mathcal{S}$ and $a \in \mathcal{A}$. The values are specified in the following cases when line 27 is run:

$$\tilde{Q}_k^r(s,a) \leftarrow \begin{cases} \tilde{Q}_k^r(s,a) & \text{if } s \in \text{Cov}(\mathcal{C}_{l+1}) \\ Q_k^r(s,a) & \text{if } s \in \text{Cov}(\mathcal{C}_l) \setminus \text{Cov}(\mathcal{C}_{l+1}) \\ \text{initial value } 0 & \text{if } s \notin \text{Cov}(\mathcal{C}_l), \end{cases}$$

$$\tilde{Q}_k^c(s,a) \leftarrow \begin{cases} \tilde{Q}_k^c(s,a) & \text{if } s \in \text{Cov}(\mathcal{C}_{l+1}) \\ Q_k^c(s,a) & \text{if } s \in \text{Cov}(\mathcal{C}_l) \setminus \text{Cov}(\mathcal{C}_{l+1}) \\ \text{initial value } 0 & \text{if } s \notin \text{Cov}(\mathcal{C}_l), \end{cases}$$

where $Q_k^r(s,a), Q_k^c(s,a)$ are the least-square estimates using the most recently extended $\mathcal{C}_l$ at that time. The dual variable $\lambda_k$ is defined in line 29. Therefore, the $\tilde{Q}_k^p(s,a)$ used in the update of policy at line 27 can be written as $\tilde{Q}_k^p(s,a) = \text{trunc}_{[0, \frac{1}{1-\gamma}]} \tilde{Q}_k^r(s,a) + \lambda_k \text{trunc}_{[0, \frac{1}{1-\gamma}]} \tilde{Q}_k^c(s,a)$.

**Lemma 1** *Whenever LSE subroutine in line 21 of Confident-NPG-CMDP is executed during a running phase $\ell = l$ for $l \in \{0, \ldots, L\}$, the least-square estimate $\tilde{Q}_k^p(s, a)$ satisfies the following condition for all iterations $k = k_\ell, \ldots, k_{\ell+1} - 1$ associated with this phase and for all $s \in \mathrm{Cov}(\mathcal{C}_\ell)$ and $a \in \mathcal{A}$,*

$$|\tilde{Q}_k^p(s, a) - q_{\pi_k', \lambda_k}^p(s, a)| \leq \epsilon' \quad \text{for all } \pi_k' \in \Pi_{\pi_k, \mathrm{Cov}(\mathcal{C}_\ell)},$$

*where $\epsilon' = (1+U)(\omega + \sqrt{\alpha}B + (\omega + \epsilon)\sqrt{\tilde{d}})$ with $\tilde{d} = \tilde{O}(d)$ and $U$ is an upper bound on the optimal Lagrange multiplier. Similarly, for initial state $s_0$, we have*

$$|\tilde{V}_k^c(s_0) - v_{\pi_k'}^c(s_0)| \leq \omega + \sqrt{\alpha}B + (\omega + \epsilon)\sqrt{\tilde{d}} \quad \text{for all } \pi_k' \in \Pi_{\pi_k, \mathrm{Cov}(\mathcal{C}_\ell)}.$$

Proof: By using the primal-dual approach, we have reduced the CMDP problem to an unconstrained problem with a single reward of the form $r_\lambda = r + \lambda c$.

Because of line 7 have executed before entering the loop and line 30 have been executed in the previous phase $\ell = l - 1$, the initial state $s_0 \in \mathrm{Cov}(C_\ell)$. If $s_0$ is in $\mathrm{Cov}(\mathcal{C}_\ell)$ for the first time (i.e. $s_0 \in \mathrm{Cov}(\mathcal{C}_\ell) \setminus \mathrm{Cov}(\mathcal{C}_{\ell+1})$ ), then the dual variable $\lambda_k$ makes a mirror descent update in line 29 using $V_k^c(s_0)$ at that time. After line 30 is executed, the core set for the next phase $\mathcal{C}_{\ell+1} = \mathcal{C}_\ell$. This means that any states, including $s_0$, that are covered by $\mathcal{C}_\ell$ are then covered by $\mathcal{C}_{\ell+1}$. By lemma 6, the initial state $s_0$ will continue to be covered by $\mathcal{C}_{\ell+1}$ for the remainder of the algorithm's execution. This implies that the dual variable $\lambda_k$ referenced in this lemma remains fixed at the value set when $s_0$ is covered by $\mathcal{C}_\ell$ for the first time and does not change thereafter for the duration of the algorithm's execution.

Then the proof of this lemma follows similar logic to lemma 11 in the single reward setting. The result of lemma 11 uses lemma 5. For lemma 5 to hold, lemma 9 is used to verify the conditions sufficient for lemma 5 to hold. For lemma 9 to hold, one of the requirement is that the behaviour policy $\pi_{k_\ell}$ and the target policy $\pi_k$ must satisfy eq. (9). In the following paragraphs, we show that eq. (9) indeed hold with appropriate changes to the parameters of interest. Then it follows that lemma 9 holds and consequently lemma 5 holds. Once all the sufficient conditions hold, by following similar logic as in lemma 11, we have the proof.

Since the policies are updated with respect to $\tilde{Q}^p$ instead of $\tilde{Q}$ of the single-reward setting, we need to make adjustment to $\eta_1, H, m, K$ to ensure $\pi_{k_\ell}$ and $\pi_k$ indeed satisfy eq. (9). First, note the value $\tilde{Q}_k^p$ for $k = 0, \ldots, K$ are in the range of $0$ and $\frac{1+U}{1-\gamma}$. The upper bound value is the result of the primary reward function taking values in the range of $[0, 1]$ and the dual variable taking values in the range of $[0, U]$. The value $U$ is defined in lemma 13 for relaxed-feasibility and in lemma 15 for strict-feasibility, and it is an upper bound on the optimal dual variable (i.e., $\lambda^* \leq U$). By similar argument to lemma 8, we make the following changes to $\eta_1, H, m, K$. We set the step size $\eta_1 = \frac{1-\gamma}{1+U}\sqrt{\frac{2\ln(|\mathcal{A}|)}{K}}$, the total number of iterations $K = \frac{6^2(\sqrt{2\ln(|\mathcal{A}|)}+1)^2(1+U)^2}{(1-\gamma)^4\epsilon^2}$, and $H = \frac{\ln((30\sqrt{\tilde{d}}(1+U))/((1-\gamma)^2\epsilon))}{1-\gamma}$. Then, it follows that $m = \frac{(1+U)\ln(1+\rho_0)}{2H\epsilon(1-\gamma)^2}$.

Next, from lemma 7, we have each $\bar{q}_k^r(s, a)$ and $\bar{q}_k^c(s, a)$ is an unbiased estimate of $\mathbb{E}_{\pi_k', s, a}[\sum_{h=0}^{H-1} \gamma^h R_{h+1}]$ and $\mathbb{E}_{\pi_k', s, a}[\sum_{h=0}^{H-1} \gamma^h C_{h+1}]$ respectively for all $\pi_k' \in \Pi_{\pi_k, \mathrm{Cov}(\mathcal{C}_l)}$. Let $\delta' = 2\exp\left(-\frac{2n\left(\frac{\epsilon}{4}\right)^2}{\left(\frac{(1+\rho_0)}{1-\gamma}\right)^2}\right)$. By lemma 9, with probability $1 - \delta'$, we have for any $(s, a) \in \mathcal{C}_l$,

$$|\bar{q}_k^r(s, a) - q_{\pi_k'}^r| \leq \epsilon, \quad |\bar{q}_k^c(s, a) - q_{\pi_k'}^c| \leq \epsilon \quad \text{for all } \pi_k' \in \Pi_{\pi_k, \mathrm{Cov}(\mathcal{C}_l)}.$$

Then the conditions of lemma 5 hold, and by similar argument to lemma 11 using lemma 5, we have for each $(s, a) \in \mathrm{Cov}(\mathcal{C}_l)$,

$$|\tilde{Q}_k^r(s, a) - q_{\pi_k'}^r(s, a)| \leq \omega + \sqrt{\alpha}B + (\omega + \epsilon)\sqrt{\tilde{d}},$$

$$|\tilde{Q}_k^c(s, a) - q_{\pi_k'}^c(s, a)| \leq \omega + \sqrt{\alpha}B + (\omega + \epsilon)\sqrt{\tilde{d}},$$

for all $\pi'_k \in \Pi_{\pi_k, \mathrm{Cov}(\mathcal{C}_\ell)}$. Then it follows that for a given $\lambda_k$,

$$|\tilde{Q}^p_k(s,a) - q^p_{\pi'_k, \lambda_k}(s,a)| = |(\tilde{Q}^r_k(s,a) - q^r_{\pi'_k}(s,a)) + \lambda_k(\tilde{Q}^c_k(s,a) - q^c_{\pi'_k}(s,a))| \leq \epsilon',$$

for all $\pi'_k \in \Pi_{\pi_k, \mathrm{Cov}(\mathcal{C}_\ell)}$.

Finally, since $s_0 \in \mathrm{Cov}(\mathcal{C}_\ell)$ and $\tilde{V}^c_k(s_0) = \langle \pi'_k(\cdot|s_0), \tilde{Q}^c_k(s_0, \cdot) \rangle$, therefore $|\tilde{V}^c_k(s_0) - v^c_{\pi_k}(s_0)| = |\langle \pi'_k(\cdot|s_0), \tilde{Q}^c_k(s_0, \cdot) - q^c_{\pi'_k}(s_0, \cdot) \rangle| \leq \omega + \sqrt{\alpha}B + (\omega + \epsilon)\sqrt{\tilde{d}}$ for all $\pi'_k \in \Pi_{\pi_k, \mathrm{Cov}(\mathcal{C}_\ell)}$. ∎

## C    Relaxed-feasibility

**Lemma 13** *[Lemma 4.1 of Jain et al. (2022)] Let $\lambda^*$ be the optimal dual variable that satisfies $\min_{\lambda \geq 0} \max_\pi v^r_\pi(\rho) + \lambda(v^c_\pi(\rho) - b)$. If we choose*

$$U = \frac{2}{\zeta(1-\gamma)},$$

*then $\lambda^* \leq U$.*

Proof: Let $\pi^*_c(\rho) = \arg\max v^c_\pi(\rho)$, and recall that $\zeta = v^c_{\pi^*_c}(\rho) - b > 0$, then

$$v^r_{\pi^*}(\rho) = \max_\pi \min_{\lambda \geq 0} v^r_\pi(\rho) + \lambda(v^c_\pi(\rho) - b).$$

By Altman (2021),

$$\begin{aligned}
v^r_{\pi^*}(\rho) &= \min_{\lambda \geq 0} \max_\pi v^r_\pi(\rho) + \lambda(v^c_\pi(\rho) - b) \\
&= \max_\pi v^r_\pi(\rho) + \lambda^*(v^c_\pi(\rho) - b) \\
&\geq v^r_{\pi^*_c}(\rho) + \lambda^*(v^c_{\pi^*_c}(\rho) - b) \\
&\geq v^r_{\pi^*_c}(\rho) + \lambda^*\zeta.
\end{aligned}$$

After rearranging terms, we have

$$\lambda^* \leq \frac{v^r_{\pi^*}(\rho) - v^r_{\pi^*_c}(\rho)}{\zeta} \leq \frac{1}{\zeta(1-\gamma)}.$$

By choosing $U = \frac{2}{\zeta(1-\gamma)}$, we have $\lambda^* \leq U$. ∎

**Definition 2**

$$R^p(\pi^*, K) = \sum_{k=0}^{K-1} \mathbb{E}_{s' \sim d_{\pi^*}(s_0), s' \in \mathrm{Cov}(\mathcal{C}_0)} \left[ \langle \pi^*(\cdot|s') - \pi_k(\cdot|s'), \tilde{Q}^r_k(s', \cdot) + \lambda_k \tilde{Q}^c_k(s', \cdot) \rangle \right],$$

$$R^d(\lambda, K) = \sum_{k=0}^{K-1} (\lambda_k - \lambda)(\tilde{V}^c_k(s_0) - b).$$

**Lemma 2** *Let $\delta \in (0,1]$ be the failure probability, $\epsilon > 0$ be the target accuracy, and $s_0$ be the initial state. Assuming for all $s \in \mathrm{Cov}(\mathcal{C}_0)$ and all $a \in \mathcal{A}$, $|\tilde{Q}^p_k(s,a) - q^p_{\pi'_k, \lambda_k}(s,a)| \leq \epsilon'$ and $|\tilde{V}^c_k(s_0) - v^c_{\pi'_k}(s_0)| \leq \omega + \sqrt{\alpha}B + (\omega + \epsilon)\sqrt{\tilde{d}}$ for all $\pi'_k \in \Pi_{\pi_k, \mathrm{Cov}(\mathcal{C}_0)}$, then, with probability $1 - \delta$, Confident-NPG-CMDP returns a mixture policy $\bar{\pi}_K$ that satisfies the following,*

$$v^r_{\pi^*}(s_0) - v^r_{\bar{\pi}_K}(s_0) \leq \frac{5\epsilon'}{1-\gamma} + \frac{(\sqrt{2\ln(|\mathcal{A}|)} + 1)(1 + U)}{(1-\gamma)^2\sqrt{K}},$$

$$b - v^c_{\bar{\pi}_K}(s_0) \leq [b - v^c_{\bar{\pi}_K}(s_0)]_+ \leq \frac{5\epsilon'}{(1-\gamma)(U - \lambda^*)} + \frac{(\sqrt{2\ln(|\mathcal{A}|)} + 1)(1 + U)}{(1-\gamma)^2(U - \lambda^*)\sqrt{K}},$$

*where $\epsilon' = (1 + U)(\omega + (\sqrt{\alpha}B + (\omega + \epsilon)\sqrt{\tilde{d}}))$ with $\tilde{d} = \tilde{O}(d)$.*

Proof: For the following result, we consider a $k \in \{0, \ldots, K\}$ with its corresponding $l \in \{0, \ldots, L\}$. At the time of termination, all $\mathcal{C}_l$ are equal.

To obtain a bound on the suboptimality and the constraint violation, we apply lemma 12 with $\pi = \pi^*$ of CMDP, $\tilde{Q}_k^p = \tilde{Q}_k^r + \lambda_k \tilde{Q}_k^c$ instead of $\tilde{Q}_k$, and lemma 1 instead of lemma 11 of the single reward setting. Then, we have

$$\frac{1}{K} \sum_{k=0}^{K-1} v_{\pi^*, \lambda_k}^p(s_0) - v_{\pi_k, \lambda_k}^p(s_0) \tag{18}$$

$$\leq \frac{4\epsilon'}{1-\gamma} + \frac{1}{K(1-\gamma)} \sum_{k=0}^{K-1} \mathbb{E}_{s' \sim d_{\pi^*}(s_0), s' \in \text{Cov}(\mathcal{C}_0)} \left[ \langle \tilde{Q}_k^r(s', \cdot) + \lambda_k \tilde{Q}_k^c(s', \cdot), \pi^*(\cdot | s') - \pi_k(\cdot | s') \rangle \right]$$

$$= \frac{4\epsilon'}{1-\gamma} + \frac{R^p(\pi^*, K)}{K(1-\gamma)}.$$

By Proposition 28.6 of Lattimore and Szepesvári (2020), the primal regret $R^p(\pi^*, K) \leq \frac{1+U}{1-\gamma} \sqrt{2K \ln(|\mathcal{A}|)}$ with $\eta_1 = \frac{1-\gamma}{1+U} \sqrt{\frac{2 \ln(|\mathcal{A}|)}{K}}$. Expanding eq. (18) in terms of $v^r, v^c$, we have

$$\frac{1}{K} \sum_{k=0}^{K-1} v_{\pi^*}^r(s_0) - v_\pi^r(s_0) + \frac{1}{K} \sum_{k=0}^{K-1} \lambda_k(v_{\pi^*}^c(s_0) - v_{\pi_k}^c(s_0))$$

$$\leq \frac{4\epsilon'}{1-\gamma} + \frac{1+U}{(1-\gamma)^2} \sqrt{\frac{2 \ln(|\mathcal{A}|)}{K}}. \tag{19}$$

Furthermore, by lemma 1, we have $|\tilde{Q}_k^c(s, a) - q_{\pi_{k'}}^c(s, a)| \leq \omega + \sqrt{\alpha} B + (\omega + \epsilon) \sqrt{\bar{d}}$ for any $s \in \text{Cov}(\mathcal{C}_l)$. Recall $\tilde{V}_k^c(s_0) = \langle \pi_k(\cdot | s_0), \tilde{Q}_k^c(s_0, \cdot) \rangle$, then it follows that $\lambda_k(v_{\pi_k}^c(s_0) - \tilde{V}_k^c(s_0)) \leq |\lambda_k(v_{\pi_k}^c(s_0) - \tilde{V}_k^c(s_0))| \leq U(\omega + \sqrt{\alpha} B + (\omega + \epsilon) \sqrt{\bar{d}}) \leq \epsilon'$.

$$\frac{1}{K} \sum_{k=0}^{K-1} \lambda_k(v_{\pi_k}^c(s_0) - v_{\pi^*}^c(s_0)) \leq \frac{1}{K} \sum_{k=0}^{K-1} \lambda_k(v_{\pi_k}^c(s_0) - b)$$

$$= \frac{1}{K} \sum_{k=0}^{K-1} \lambda_k(v_{\pi_k}^c(s_0) - \tilde{V}_k^c(s_0)) + \lambda_k(\tilde{V}_k^c(s_0) - b)$$

$$\leq \epsilon' + \frac{R^d(0, K)}{K}$$

$$\leq \epsilon' + \frac{U}{(1-\gamma)\sqrt{K}}.$$

The update to the dual variable is a mirror descent algorithm. By Proposition 28.6 of Lattimore and Szepesvári (2020), the dual regret $R^d(0, K) \leq \frac{U\sqrt{K}}{1-\gamma}$ with $\eta_2 = \frac{U(1-\gamma)}{\sqrt{K}}$. Altogether,

$$\frac{1}{K} \sum_{k=0}^{K-1} v_{\pi^*}^r(s_0) - v_{\pi_k}^r(s_0) \leq \frac{4\epsilon'}{1-\gamma} + \frac{1+U}{(1-\gamma)^2} \sqrt{\frac{2 \ln(|\mathcal{A}|)}{K}} + \epsilon' + \frac{U}{(1-\gamma)\sqrt{K}}$$

$$\leq \frac{5\epsilon'}{1-\gamma} + \frac{(\sqrt{2 \ln(|\mathcal{A}|)} + 1)(1+U)}{(1-\gamma)^2 \sqrt{K}}$$

For bounding the constraint violations, we first incorporate $R^d(\lambda, K)$ into eq. (19) and rearrange terms to obtain:

$$\frac{1}{K}\sum_{k=0}^{K-1} v^r_{\pi^*}(s_0) - v^r_{\pi_k}(s_0) + \frac{\lambda}{K}\sum_{k=0}^{K-1}(b - v^c_{\pi_k}(s_0))$$

$$\leq \frac{1}{K}\sum_{k=0}^{K-1}(\lambda_k - \lambda)(v^c_{\pi_k}(s_0) - b) + \frac{4\epsilon'}{1-\gamma} + \frac{(1+U)\sqrt{2\ln(|\mathcal{A}|)}}{(1-\gamma)^2\sqrt{K}}$$

$$= \frac{1}{K}\sum_{k=0}^{K-1}(\lambda_k - \lambda)(v^c_{\pi_k}(s_0) - \tilde{V}^c_k(s_0)) + \frac{1}{K}\sum_{k=0}^{K-1}(\lambda_k - \lambda)(\tilde{V}^c_k(s_0) - b)$$

$$+ \frac{4\epsilon'}{1-\gamma} + \frac{(1+U)\sqrt{2\ln(|\mathcal{A}|)}}{(1-\gamma)^2\sqrt{K}}$$

$$= \epsilon' + \frac{R^d(\lambda, K)}{K} + \frac{4\epsilon'}{1-\gamma} + \frac{(1+U)\sqrt{2\ln(|\mathcal{A}|)}}{(1-\gamma)^2\sqrt{K}}$$

$$\leq \frac{5\epsilon'}{1-\gamma} + \frac{(1+U)(\sqrt{2\ln(|\mathcal{A}|)}+1)}{(1-\gamma)^2\sqrt{K}}$$

There are two constraint cases. Case one is no violation: $b - v^c_{\bar{\pi}_K}(s_0) \leq 0$. Then, it also holds that $b - \triangle - v^c_{\bar{\pi}_K}(s_0) \leq 0$ for any $\triangle \geq 0$, which is what we want to show. Case two is violation: $b - v^c_{\bar{\pi}_K}(s_0) > 0$, for which case, $\lambda = U$. Using notation $[x]_+ = \max\{x, 0\}$, we have

$$\frac{1}{K}\sum_{k=0}^{K-1} v^r_{\pi^*}(s_0) - v^r_{\pi_k}(s_0) + \frac{U}{K}\left[\sum_{k=0}^{K} b - v^c_{\pi}(s_0)\right]_+$$

$$\leq \frac{5\epsilon'}{1-\gamma} + \frac{(1+U)(\sqrt{2\ln(|\mathcal{A}|)}+1)}{(1-\gamma)^2\sqrt{K}}.$$

By Lemma B.2 of Jain et al. (2022), we have

$$[b - v^c_{\bar{\pi}_K}(s_0)]_+ \leq \frac{5\epsilon'}{(1-\gamma)(U-\lambda^*)} + \frac{(\sqrt{2\ln(|\mathcal{A}|)}+1)(1+U)}{(1-\gamma)^2(U-\lambda^*)\sqrt{K}}.$$

∎

**Theorem 1** *With probability $1-\delta$, the mixture policy $\bar{\pi}_K$ returned by confident-NPG-CMDP ensures that*

$$v^r_{\pi^*}(s_0) - v^r_{\bar{\pi}_K}(s_0) = \frac{5(1+U)(1+\sqrt{\tilde{d}})}{1-\gamma}\omega + \epsilon, \tag{20}$$

$$v^c_{\bar{\pi}_K}(s_0) \geq b - \left(\frac{5(1+U)(1+\sqrt{\tilde{d}})}{(1-\gamma)}\omega + \epsilon\right). \tag{21}$$

*if we choose $n = \frac{30^2(1+\rho_0)^2(1+U)^2\tilde{d}}{2\epsilon^2(1-\gamma)^4}\ln\left(\frac{8\tilde{d}(L+1)}{\delta}\right)$, $\alpha = \frac{(1-\gamma)^2\epsilon^2}{30^2(1+U)^2B^2}$, $K = \frac{6^2(\sqrt{2\ln(|\mathcal{A}|)}+1)^2(1+U)^2}{(1-\gamma)^4\epsilon^2}$, $\eta_1 = \frac{1-\gamma}{1+U}\sqrt{\frac{2\ln(|\mathcal{A}|)}{K}}$, $\eta_2 = \frac{U(1-\gamma)}{\sqrt{K}}$, $H = \frac{\ln((30\sqrt{\tilde{d}}(1+U))/((1-\gamma)^2\epsilon))}{1-\gamma}$, $m = \frac{(1+U)\ln(1+\rho_0)}{2\epsilon H(1-\gamma)^2}$, and $U = \frac{2}{\zeta(1-\gamma)}$.*

*Furthermore, the algorithm utilizes at most $\tilde{O}(d^2(1 + U)^3\epsilon^{-3}(1 - \gamma)^{-8})$ queries in the local-access setting.*

Proof: From lemma 2, we have

$$v^r_{\pi^*}(s_0) - v^r_{\bar{\pi}_K}(s_0) \leq \frac{5\epsilon'}{(1-\gamma)} + \frac{(\sqrt{2\ln(|\mathcal{A}|)}+1)(1+U)}{(1-\gamma)^2\sqrt{K}}, \tag{22}$$

$$b - v^c_{\bar{\pi}_K}(s_0) \leq \frac{5\epsilon'}{(1-\gamma)(U-\lambda^*)} + \frac{(\sqrt{2\ln(|\mathcal{A}|)}+1)(1+U)}{(1-\gamma)^2(U-\lambda^*)\sqrt{K}}, \tag{23}$$

Let $C = \frac{1}{\zeta(1-\gamma)}$ for a $\zeta \in (0, \frac{1}{1-\gamma}]$. By lemma 13, we chose $U = 2C$ and $\lambda^* \leq C$. It follows that $\frac{1}{U-\lambda^*} \leq \frac{1}{C} = \zeta(1-\gamma) \leq 1$, and thus the right hand side of eq. (23) is upper bounded by the right hand side of eq. (22). Recall $\epsilon' = (1+U)\left(\omega + \left(\sqrt{\alpha}B + (\omega + \epsilon)\sqrt{\tilde{d}}\right)\right)$. Then, the goal is to set the parameters $H, n, K$, and $\alpha$ appropriately so that the $A, B$ and $C$ of the following expression, when added together, is less than $\epsilon$:

$$\frac{5(1+U)(1+\sqrt{\tilde{d}})\omega}{1-\gamma} + \underbrace{\frac{5(1+U)\sqrt{\alpha}B}{1-\gamma}}_{A} + \underbrace{\frac{5(1+U)\epsilon\sqrt{\tilde{d}}}{1-\gamma}}_{B} + \underbrace{\frac{(\sqrt{2\ln(|\mathcal{A}|)}+1))(1+U)}{(1-\gamma)^2\sqrt{K}}}_{C} . \quad (24)$$

First, we set $n$ appropriately so that the failure probability is well controlled. The failure probability depends on the number of times Gather-data subroutine (algorithm 3) is executed. Gather-data is run for phase $0, \ldots, L$. Each phase has at most $\tilde{d}$ elements, and recall $\tilde{d}$ is defined in lemma 10. Therefore, Gather-data would return success at most $\tilde{d}$ times. Altogether, Gather-data can return success at most $\tilde{d}(L+1)$ times, each with probability of at least $1 - \delta' = 1 - \delta/(\tilde{d}(L+1))$. By a union bound, Gather-data returns success in all occasions with probability $1 - \delta$.

By setting $H = \frac{\ln((30\sqrt{\tilde{d}}(1+U))/((1-\gamma)^2\epsilon))}{1-\gamma}$ and $n = \frac{30^2(1+\rho_0)^2(1+U)^2\tilde{d}}{2\epsilon^2(1-\gamma)^4} \ln\left(\frac{8\tilde{d}(L+1)}{\delta}\right)$, we have for any $l \in \{0, \ldots, L\}$, $k = k_l, \ldots, k_{l+1} - 1$, the $|\bar{q}_k^r(s,a) - q_{\pi_k'}^r(s,a)| \leq \frac{4}{6}\frac{(1-\gamma)\epsilon}{5(1+U)\sqrt{\tilde{d}}}$ and $|\bar{q}_k^c(s,a) - q_{\pi_k'}^c(s,a)| \leq \frac{4}{6}\frac{(1-\gamma)\epsilon}{5(1+U)\sqrt{\tilde{d}}}$ hold for all $\pi_k' \in \Pi_{\pi_k, \text{Cov}(\mathcal{C}_l)}$ with probability at least $1 - \delta$. Then, this is used in the accuracy guarantee of the least-square estimate (lemma 1) and finally in the suboptimality bound of lemma 2.

Then, we can set $\alpha$ of eq. (24) to be equal to $\frac{\epsilon}{6}$ and solve for $\alpha = \frac{\epsilon^2(1-\gamma)^2}{30^2(1+U)^2B^2}$. Finally, by setting $K = \frac{6^2(\sqrt{2\ln(|\mathcal{A}|)}+1)^2(1+U)^2}{(1-\gamma)^4\epsilon^2}$, we have $C$ of eq. (24) be less than $\frac{\epsilon}{6}$. Altogether, we have the reward suboptimality satisfying eq. (20) and constraint satisfying eq. (21).

For the query complexity, we note that our algorithm does not query the simulator in every iteration, but at fixed intervals, which we call phases. Each phase is $m$ iterations in length. There are total of $L = \lfloor K/(\lfloor m \rfloor + 1) \rfloor \leq K/m = \tilde{O}\left((1+U)(1-\gamma)^{-3}\epsilon^{-1}\right)$ phases. In each phases, Gather-data subroutine (algorithm 3) can be run. Each time Gather-data returns success with trajectories, the subroutine would have made at most $nH$ queries. Gather-data is run for each of the elements in $\mathcal{C}_l$, $l \in \{0, \ldots, L\}$. By the time the algorithm terminates, all $\mathcal{C}_l$'s are the same. Since there are at most $\tilde{O}(d)$ elements in each $\mathcal{C}_l$, the algorithm will make a total of $nH(L+1)|\mathcal{C}_0|$ number of queries to the simulator. Since we have $H = \tilde{O}((1-\gamma)^{-1})$, $n = \tilde{O}((1+U)^2 d\epsilon^{-2}(1-\gamma)^{-4})$ and $L = \tilde{O}((1+U)\epsilon^{-1}(1-\gamma)^{-3})$, the sample complexity is $\tilde{O}(d^2(1+U)^3(1-\gamma)^{-8}\epsilon^{-3})$. ∎

## D Strict-feasibility

**Lemma 14** *Let $\pi^*_\triangle$ be defined as in eq. (7) and $\pi^*$ be an optimal policy of CMDP. Then, for a $\triangle > 0$,*

$$v_{\pi^*}^r(s_0) - v_{\pi^*_\triangle}^r(s_0) \leq \lambda^*\triangle,$$

*where $\lambda^*$ is an optimal dual variable that satisfies $\min_{\lambda \geq 0} \max_\pi v_\pi^r(s_0) + \lambda(v_\pi^c(s_0) - b')$.*

Proof:

$$v_{\pi^*_\triangle}^r(s_0) = \max_\pi \min_{\lambda \geq 0} v_\pi^r(s_0) + \lambda(v_\pi^c(s_0) - b').$$

By Altman (2021),

$$v^r_{\pi^*_\triangle}(s_0) = \min_{\lambda \geq 0} \max_\pi v^r_\pi(s_0) + \lambda(v^c_\pi(s_0) - b')$$

$$= \max_\pi v^r_\pi(s_0) + \lambda^*(v^c_\pi(s_0) - b')$$

$$\geq v^r_{\pi^*}(s_0) + \lambda^*(v^c_{\pi^*}(s_0) - (b + \triangle))$$

$$\geq v^r_{\pi^*}(s_0) + \lambda^*(b - b - \triangle) \quad \text{because } v^c_{\pi^*}(s_0) \geq b$$

$$= v^r_{\pi^*}(s_0) - \lambda^*\triangle.$$

After rearranging the terms, we get the result. ∎

**Lemma 15** *Let $\lambda^*$ be the optimal dual variable that satisfies $min_{\lambda \geq 0} \max_\pi V^r_\pi(s_0) + \lambda(V^c_\pi(s_0) - b')$. If we choose*

$$U = \frac{4}{\zeta(1-\gamma)},$$

*then $\lambda^* \leq U$ requiring that $\triangle \in (0, \frac{\zeta}{2})$.*

Proof: Let $\pi^*_c(s_0) = \arg \max V^c_\pi(s_0)$, and recall that $\zeta = V^c_{\pi^*_c}(s_0) - b > 0$, then

$$v^r_{\pi^*_\triangle}(s_0) = \max_\pi \min_{\lambda \geq 0} v^r_\pi(s_0) + \lambda(v^c_\pi(s_0) - b')$$

By Altman (2021),

$$v^r_{\pi^*_\triangle}(s_0) = \min_{\lambda \geq 0} \max_\pi v^r_\pi(s_0) + \lambda(v^c_\pi(s_0) - b')$$

$$= \max_\pi v^r_\pi(s_0) + \lambda^*(V^c_\pi(s_0) - b')$$

$$\geq v^r_{\pi^*_c}(s_0) + \lambda^*(v^c_{\pi^*_c}(s_0) - (b + \triangle))$$

$$= v^r_{\pi^*_c}(s_0) + \lambda^*(\zeta - \triangle).$$

If we require $\triangle \in (0, \frac{\zeta}{2})$, then we have

$$v^r_{\pi^*_\triangle}(s_0) \geq v^r_{\pi^*_c}(s_0) + \lambda^*(\zeta - \frac{\zeta}{2})$$

$$= v^r_{\pi^*_c}(s_0) + \frac{\lambda^*\zeta}{2} \qquad (25)$$

After rearranging terms in eq. (25), we have

$$\lambda^* \leq \frac{2(v^r_{\pi^*_\triangle}(s_0) - v^r_{\pi^*_c}(s_0))}{\zeta} \leq \frac{2}{\zeta(1-\gamma)}.$$

By choosing $U = \frac{4}{\zeta(1-\gamma)}, \lambda^* \leq U$. ∎

**Theorem 2** *With probability $1 - \delta$, a target $\epsilon > 0$, the mixture policy $\bar{\pi}_K$ returned by confident-NPG-CMDP ensures that $v^r_{\pi^*}(s_0) - v^r_{\bar{\pi}_K}(s_0) \leq \epsilon$ and $v^c_{\bar{\pi}_K}(s_0) \geq b$, if assuming the misspecification error $\omega \leq \frac{\triangle(1-\gamma)}{70(1+U)(1+\sqrt{\bar{d}})}$, and if we choose $\triangle = \frac{\epsilon(1-\gamma)\zeta}{8}, \alpha = \frac{\triangle^2(1-\gamma)^2}{70^2(1+U)^2B^2}, K = \frac{14^2(\sqrt{2\ln(|\mathcal{A}|)}+1)^2(1+U)^2}{(1-\gamma)^4\triangle^2}, n = \frac{(14*5)^2(1+\rho_0)^2\bar{d}(1+U)^2}{2\triangle^2(1-\gamma)^4} \ln\left(\frac{8\bar{d}(L+1)}{\delta}\right), H = \frac{\ln\left(\frac{14*5(1+U)\sqrt{\bar{d}}}{\triangle(1-\gamma)^2}\right)}{1-\gamma}, m = \frac{(1+U)\ln(1+\rho_0)}{2\triangle H(1-\gamma)^2}, U = \frac{4}{\zeta(1-\gamma)}.$*

*Furthermore, the algorithm utilizes at most $\tilde{O}(d^2(1 + U)^3(1-\gamma)^{-11}\epsilon^{-3}\zeta^{-3})$ queries in the local-access setting.*

Proof: Let $\lambda^*$ be the optimal dual variable that satisfies the Lagrangian primal-dual of the surrogate CMDP defined by eq. (7) (i.e., $\lambda^* = \arg \min_{\lambda \geq 0} \max_\pi v^r_\pi(s_0) + \lambda(v^c_\pi(s_0) - b')$).

$$v^r_{\pi^*}(s_0) - v^r_{\bar{\pi}_K}(s_0)$$

$$= \underbrace{\left[v^r_{\pi^*}(s_0) - v^r_{\pi^*_\triangle}(s_0)\right]}_{\text{surrogate suboptimality}} + \underbrace{\left[v^r_{\pi^*_\triangle}(s_0) - v^r_{\bar{\pi}_K}(s_0)\right]}_{\text{Confident-NPG-CMDP suboptimality}}$$

$$\leq \lambda^*\triangle + \bar{\epsilon},$$

where $\bar{\epsilon} = \frac{5(1+U)(1+\sqrt{\tilde{d}})\omega}{1-\gamma} + \frac{5(1+U)\sqrt{\alpha}B}{1-\gamma} + \frac{5(1+U)\epsilon\sqrt{\tilde{d}}}{1-\gamma} + \frac{(\sqrt{2\ln(|\mathcal{A}|)}+1)(1+U)}{(1-\gamma)^2\sqrt{K}}$. By lemma 14, $v_{\pi^*}^r(s_0) - v_{\pi^*_\triangle}^r(s_0) \leq \lambda^*\triangle$. We can further upper bound $\lambda^*$ by $U = \frac{4}{\zeta(1-\gamma)}$ using lemma 15 and requiring $\triangle \in \left(0, \frac{\zeta}{2}\right)$. Together with theorem 1, we have Confident-NPG-CMDP return $\bar{\pi}_K$ s.t.

$$v_{\pi^*}^r(s_0) - v_{\bar{\pi}_K}^r(s_0) \leq \frac{4\triangle}{\zeta(1-\gamma)} + \bar{\epsilon} \quad \text{and} \tag{26}$$

$$b' - V_{\bar{\pi}_K}^c(s_0) \leq \bar{\epsilon}.$$

Now, we need to set $\triangle$ such that 1) $\triangle \in \left(0, \frac{\zeta}{2}\right)$ and 2) $\triangle - \bar{\epsilon} \geq 0$ are satisfied. If we choose $\triangle = \frac{\epsilon(1-\gamma)\zeta}{8}$, then the first condition is satisfied. This is because $\epsilon \in \left(0, \frac{1}{1-\gamma}\right]$, and thus $\triangle \leq \frac{\zeta}{8} < \frac{\zeta}{2}$.

Next, we check if our choice of $\triangle = \frac{\epsilon(1-\gamma)\zeta}{8}$ satisfies $\triangle - \bar{\epsilon} \geq 0$. For the condition $\triangle - \bar{\epsilon} \geq 0$ to be true, we make an assumption on the misspecification error $\omega \leq \frac{\triangle(1-\gamma)}{70(1+U)(1+\sqrt{\tilde{d}})}$, and pick $n, \alpha, K, \eta_1, \eta_2, H, m$ to be the values outlined in this theorem. Consequently, we have $\bar{\epsilon} = \frac{1}{2}\triangle$. Then, we have ensured the condition $\triangle - \bar{\epsilon} \geq 0$ is satisfied.

We note that because $\zeta \in \left(0, \frac{1}{1-\gamma}\right)$, we have $\bar{\epsilon} \leq \frac{\epsilon}{16} \leq \epsilon$. following from eq. (26), we have $v_{\pi^*}^r(s_0) - v_{\bar{\pi}_K}^r(s_0) \leq \epsilon$ and $b' - V_{\bar{\pi}_K}^c(s_0) \leq \frac{\triangle}{2}$. Then it follows that $b + \frac{\triangle}{2} \leq V_{\bar{\pi}_K}^c(s_0)$. Strict-feasilbility is achieved.

For the query complexity, we note that our algorithm does not query the simulator in every iteration, but at fixed intervals, which we call phases. Each phase is $m$ iterations in length. There are total of $L = \lfloor K/(\lfloor m \rfloor + 1) \rfloor \leq K/m = \tilde{O}\left((1+U)(1-\gamma)^{-3}\triangle^{-1}\right)$ phases. In each phase, Gather-data subroutine (algorithm 3) can be run. Each time Gather-data subroutine returns with trajectories, the subroutine would have made at most $nH$ queries. Gather-data is run for each of the element in $\mathcal{C}_l$, $l \in \{0, \ldots, L\}$. By the time the algorithm terminates, all $\mathcal{C}_l$'s are the same. Since there are at most $\tilde{O}(d)$ elements in each $\mathcal{C}_l$, the algorithm will make a total of $nH(L+1)|\mathcal{C}_0|$ number of queries to the simulator. Since we have $H = \tilde{O}((1-\gamma)^{-1})$, $n = \tilde{O}((1+U)^2 d(1-\gamma)^{-4}\triangle^{-2})$, $L = \tilde{O}\left((1+U)(1-\gamma)^{-3}\triangle^{-1}\right)$, and $\triangle = \frac{\epsilon\zeta(1-\gamma)}{8}$, the sample complexity is $\tilde{O}(d^2(1+U)^3(1-\gamma)^{-11}\epsilon^{-3}\zeta^{-3})$. $\blacksquare$

# E    A discussion on memory cost and some implementation details

By recording the states added to each core set during extensions and their corresponding least-squares weights, we can reconstruct the policy as needed. This section explains how to track this information and how it facilitates policy reconstruction.

In phase $l$, the policies $\pi_k$ for iterations $k = k_l + 1, \ldots, k_{l+1} - 1$ depend on the core set $\mathcal{C}_l$. Since $\mathcal{C}_l$ can be extended multiple times, these policies may change accordingly. However, we do not want to change the action distribution for states have already passed the uncertainty test in previous extensions (i.e. $s \in \text{Cov}(\mathcal{C}_{l+1})$). For such states, action distributions are based on the least-square estimation of the core set at that time they passed the uncertainty test for the first time (i.e., $s \in \text{Cov}(\mathcal{C}_l) \setminus \text{Cov}(\mathcal{C}_{l+1})$). Therefore, it is essential to track newly added states in each extension and store their corresponding least-square weights to recompute their action distributions.

To achieve this, $\mathcal{C}_0$ is extended only via line 7 and line 15 of algorithm 1, while other core sets $\mathcal{C}_l$, where $l \in \{1, \ldots, L+1\}$ are extended solely via line 30 during the running phase $\ell = l - 1$. We mark newly added elements in line 7, line 15, and line 30. After executing line 21, we store the least-square weights associated with these newly added state-action pairs.

By keeping track of the state-action pairs that are newly added in each extension and saving the corresponding least-square weights, we can construct the policy $\pi_{k+1}$ associated with $\mathcal{C}_l$. Let $\mathcal{C}_l^0 = \emptyset$, and $\mathcal{C}_l^i$ denote all state-action pairs added to $\mathcal{C}_l$ in extension $i$ for $i = 1$ up to at most $\tilde{d}$. Let $w_k^i$ represent the least-square weight computed using $\mathcal{C}_l = \mathcal{C}_l^0 \cup \mathcal{C}_l^1 \cup \mathcal{C}_l^2 \cup \cdots \cup \mathcal{C}_l^i$ for the $k$-th iteration. When $\mathcal{C}_l$ is extended for the $(i+1)$-th time, let $\mathcal{C}_l^{i+1}$ be the set of newly added state-action pairs,

making the latest $\mathcal{C}_l = \mathcal{C}_l^0 \cup \mathcal{C}_l^1 \cup \mathcal{C}_l^2 \cup \cdots \cup \mathcal{C}_l^{i+1}$. The least-squares weight $w_k^{i+1}$ is then computed using $\mathcal{C}_l$. When line 27 of algorithm 1 is executed, $\pi_{k+1}$ remains unchanged for the rest of the algorithm's execution for any states already in $\mathrm{Cov}(\mathcal{C}_l^0 \cup \mathcal{C}_l^1 \cup \cdots \cup \mathcal{C}_l^i)$, equivalent to $\mathrm{Cov}(\mathcal{C}_{l+1})$ in line 27, because line 30 would have been executed in the $i$-th extension, making $\mathcal{C}_{l+1} = \mathcal{C}_l^0 \cup \mathcal{C}_l^1 \cup \cdots \cup \mathcal{C}_l^i$. For states in $\mathrm{Cov}(\mathcal{C}_l^0 \cup \mathcal{C}_l^1 \cup \cdots \cup \mathcal{C}_l^{i+1}) \setminus \mathrm{Cov}(\mathcal{C}_l^0 \cup \mathcal{C}_l^1 \cup \cdots \cup \mathcal{C}_l^i)$ (equivalent to $\mathrm{Cov}(\mathcal{C}_l) \setminus \mathrm{Cov}(\mathcal{C}_{l+1})$ in line 27), $\pi_{k+1}$ makes a softmax update using $w_k^{i+1}$. For all other states not in $\mathrm{Cov}(\mathcal{C}_l)$, the policy remains as $\pi_k$.

A subroutine can start with $\pi_0$, use the stored data to compute and return $\pi_k(\cdot|s)$ for any $s$ and $k$. By tracking newly added elements and the corresponding least-square weights, the algorithm can reconstruct policies $\pi_0, \ldots, \pi_K$. This approach enables the algorithm to return the value of a mixture policy at termination.

