# OpenReview forum: "Confident Natural Policy Gradient for Local Planning in  $q_\pi$-realizable Constrained MDPs"
_NeurIPS.cc/2024/Conference — NeurIPS 2024 poster_

### Official Review · Reviewer_KqmD · 2024-07-05

**Soundness:** 3
**Presentation:** 3
**Contribution:** 2
**Rating:** 6
**Confidence:** 2

**Summary:**

This is the first work which addresses and achieves polynomial sample complexity for the learning problem of CMDPs in the more general setting of linear function approximation with $q_{\pi}$ realizability.  The authors propose a primal-dual algorithm and utilize a local access model (can be viewed to in between online learning and a generative model ). The algorithm reliably produces a policy that strictly adheres to the constraints while closely optimizing the reward function's value.

**Strengths:**

- The paper is mostly well written with good summarization of the main ideas and theorem statements.
- It is the first work achieving polynomial sample complexity for CMDPs in the more general $q_{\pi}$ realizability setting.

**Weaknesses:**

- Lack of experimental results.
-  The paper seems an extension of earlier work i.e., (Weisz et al., 2022).
- Despite general good writing, the main algorithm description in section 4.2-4.3 is very dense and difficult to parse.

**Questions:**

- Since the paper claims that $q_{\pi}$ realizability is more general than the linear MDP setting, does it mean that linear MDP setting implies $q_{\pi}$ realizability ? Is this result known or trivial ? Could this point be included in the paper?

- The paper seems an extension of earlier work i.e., (Weisz et al., 2022). Can the authors elaborate on what were the main technical challenges overcome on extending this work to the CMDP setting ?

**Limitations:**

There are no potential negative societal impact of their work

---

> ### Author Rebuttal · Authors · 2024-08-05
>
> Yes, you are correct that linear MDP implies $q_\pi$ realizability.  However, $q_\pi$ realizability DOES NOT imply linear MDP, and one can show this via a counterexample.  To name a few references that discuss this, please refer to Proposition 4 of Andrea Zanette et al., Learning near optimal policies with low inherent Bellman error, 2020a. (https://arxiv.org/pdf/2003.00153) and Hao, B et al., Confident least square value iteration with local access to a simulator (2022) (https://proceedings.mlr.press/v151/hao22a/hao22a.pdf).

---

### Official Review · Reviewer_EJjQ · 2024-07-06

**Soundness:** 3
**Presentation:** 2
**Contribution:** 2
**Rating:** 4
**Confidence:** 2

**Summary:**

This paper studied constrained Markov decision processes (CMDP) and proposed a confident policy gradient algorithm for $q_\pi$ realizable MDPs. Here, a $q_\pi$ realizable MDP assumes that the Q function can be approximated by a linear function w.r.t. some feature of state-action pairs. By using primal-dual methods, the proposed algorithm can find an $\epsilon$-optimal policy satisfying the constraint or with small constraint violation. The proposed approach can also be applied to the misspecification case.

**Strengths:**

1. The paper is technically sound.
2. The studied problem is important and well-motivated.

**Weaknesses:**

1. The major weakness is the presentation of contribution. According to the paper's presentation, it feels like the proposed method is a straightforward extension of Weisz et al., 2022 applying to the Lagrangian objective function. It would be better to highlight the difficulties of handling CMDP or this Lagrangian function.

2. The presentation of the two learning goals is also confusing. In my understanding, the proposed algorithm can already achieve the stringent constraint, i.e. zero violation, and the relaxed feasibility is satisfied naturally. It seems redundant to spend a whole section to describe the result of relaxed feasibility. Some papers split the two results (Liu et al., 2021) because they need different assumptions. Perhaps the authors can explain or organize the presentation more clear.

3. Another presentation issue: There is no introduction or justification of the terminology *natural policy gradient* in this paper.

[1]. Weisz et al., Confident Approximate Policy Iteration for Efficient Local Planning in $q_\pi$-realizable MDPs, 2022.

[2]. Liu et al., Learning policies with zero or bounded constraint violation for constrained mdps, 2021

**Questions:**

Please see the weaknesses part.

**Limitations:**

No further limitations need to be addressed.

---

> ### Author Rebuttal · Authors · 2024-08-05
>
> Please note that our algorithm is NOT a straightforward extension of CAPI-QPI-Plan. Moreover, our paper distinguishes between the relaxed-feasibility and strict-feasibility problem settings. Treating each setting uniquely is a key strength of our work. In the relaxed-feasibility problem, the returned policy $\bar \pi_K$ is allowed to have some constraint violations, specifically where $v_{\bar \pi_K}^c(s_0) \geq b - \epsilon$. This contrasts with the strict-feasibility setting, where no constraint violations are permitted, meaning $v_{\bar \pi_K}^c(s_0) \geq b$. To address the strict-feasibility problem, the algorithm must solve a more conservative CMDP, as discussed in Section 6 of our paper. However, solving this more conservative CMDP incurs a higher sample complexity cost, necessitating that the relaxed-feasibility setting be treated separately. Additionally, in the presence of a misspecification error $\omega > 0$, the strict-feasibility setting requires additional assumptions on $\omega$, whereas the relaxed-feasibility setting does not. The sample complexity of the relaxed-feasibility setting can be independent of Slater's constant, whereas for strict feasibility, the returned policy must strictly adhere to constraints, and we cannot simply set Slater's constant $\zeta$ to $\epsilon$ and disregard its impact.

---

> > ### Comment · Reviewer_EJjQ · 2024-08-12
> >
> > I thank the authors for the response, which largely addresses my concerns such as the separation of relaxed and strict constraint cases and differences from previous works. However, I believe that these discussions are necessary and the current version would benefit from a careful revision. I decide to slightly change my score and lower my confidence.

---

> ### Author Response · Authors · 2024-08-13
>
> We are pleased to have addressed your concerns, and we sincerely thank the reviewer for their response.

---

### Official Review · Reviewer_ynfZ · 2024-07-09

**Soundness:** 3
**Presentation:** 3
**Contribution:** 3
**Rating:** 7
**Confidence:** 3

**Summary:**

**Problem setup**

The authors consider the task of global planning in large Constrained
Discounted MDPs. They assume local access to a simulator which can be queried at previously
encountered state-action pairs to obtain a next state sample and immediate
reward, and that the $Q$-value of any policy is linearly realizable by a given feature map. Under this more general assumption on the MDP, they aim to develop an efficient method which learns a policy that maximizes return with minimal constraint violations.

**Approach**

In the above context, the authors propose a primal-dual approach to solve a constrained
return optimization problem. Under the weaker assumption of $Q^{\pi}$-realizability and local access to a simulator, their method combines existing techniques on $Q$-value estimation in large MDPs and off-policy evaluation to nicely approximate the gradients of the primal-dual objective while conserving samples. Precisely, they apply the $Q$-values estimation sub-routine of CAPI-QPI-PLAN in [1] to construct least squares estimates which are guaranteed to closely approximate the true value at a subset of important (namely \emph{core}) state-action pairs. Furthermore, contrary to the fully on-policy evaluation routine of CAPI-QPI-PLAN, their method Confident-NPG-CMDP reuses the core set and trajectory data for value estimation and policy improvement across a number of steps in each learning phase.

**Result**

The authors derive performance guarantees for their method in terms of the value error and constraint violation of the output mixture policy. Precisely, they prove that w.h.p their method learns a near-optimal policy with minimal constraint violation with at most $\mathcal{O}(poly(d)\varepsilon^{-3})$ queries to the simulator and iterations.



[1] Weisz, G., György, A., Kozuno, T., & Szepesvári, C. (2022). Confident Approximate Policy Iteration for Efficient Local Planning in $ q^\pi $-realizable MDPs. Advances in Neural Information Processing Systems, 35, 25547-25559.


**Update**: I also updated my score for contribution.

**Strengths:**

1. This paper addresses the task of planning in large Constrained Discounted MDPs under the weakest possible assumptions in the RL literature. This is achieved with a unique combination of existing techniques. Their value estimation scheme makes use of techniques from CAPI-QPI-PLAN to create a core set in addition to off-policy evaluation via importance sampling for value estimation at core state-action pairs. The authors also adequately cite related works.

2. The submission is well written and, to the best of my knowledge, technically sound.

3. In terms of the target accuracy and feature dimension, the results are significant especially in restrictive the setting of approximate policy completeness and local access to the simulator that they consider.

**Weaknesses:**

1. The relevance and impact on the theoretical analysis of the off-policy evaluation scheme is not fully accessible. It would help if the authors could clarify (with pointers in the main text), how this routine helps conserve samples in theory. Precisely, does the sample complexity get worse if the authors simply apply CAPI-QPI-PLAN?

2. In Line 268, it should be $\frac{1}{K}$ not $\frac{1}{k}$ in the definition of $\overline{\pi}_{K}$.

**Questions:**

1. The nature of the output policy in Line 268 is a bit ambiguous and I spent a considerable amount of time trying to understand the meaning of the notation $\frac{1}{K}\sum_{k=0}^{K-1}\pi_{k}$ means.

    On one hand, the authors refer to the said notation as a mixture policy, which to my knowledge means that at deployment, the agent flips a coin at the beginning of each episode to decide which of the $K$ policies to use. In this case, it is unclear how you define $v_{\overline{\pi}\_{K}}$, and more importantly, such policies are not suitable for constrained MDPs, especially with safety constraints as they are only required to perform well on average.

    On the other hand, the notation seems to mean that $\overline{\pi}\_{K}$ is an element-wise average for all state-action pairs. In this case, the policy is stochastic and $v_{\overline{\pi}\_{K}}$ well defined.

2. The complexity guarantees has worse dependence on the effective horizon $(1-\gamma)$ e.g. $(1-\gamma)^{-11}$ in Theorem 1 and $(1-\gamma)^{-14}$  in Theorem 2. Is this a consequence of the method or an artefact of the analysis? Can this be improved?

3. In Remark 3, the authors mention that the Slater's constant, which is required by their algorithm, can be approximated by another algorithm. Is this approximation trivial? How will the approximation error influence the current results?

4. In appendix A, the authors provide Confident-NPG, a version of
Confident-NPG-CMDP for the single-reward (or unconstrained MDP) case.
Furthermore, their performance guarantees for the latter algorithm appear to be
based on first relating the former to CAPI-QPI-PLAN. Therefore, to
better understand the contributions of this paper in the constrained case, the following questions are based on Confident-NPG. Can the authors kindly clarify the following:

    - Why is $c > 0$ interesting to study?

    - What is a suitable choice of $c$ and how does tuning this parameter show up in the performance guarantees?

    - Can the authors comment on the memory requirement of Confident-NPG?

**Limitations:**

This is a purely theory paper so there are no direct
societal impacts. However, I would recommend that the authors highlight the
dependence on the effective horizon in their discussion. More
importantly, I recommend that the authors elaborate on the memory and
computational complexity as these contribute to the efficiency of their method.

---

> ### Author Rebuttal · Authors · 2024-08-05
>
> You are correct that the mixture policy randomly selects an index $I \in {0, \ldots, K-1}$ with probability $1/K$ at deployment and then follows the policy $\pi_I$ for all subsequent steps. The value function of the mixture policy, $v_{\bar \pi_K}(s)$, is defined as the expected return when the mixture policy is initiated in state $s$: $v_{\bar \pi_K}(s) = \sum_{i=0}^{K-1} \text{Prob}(I = i) v_{\pi_i}(s) = \frac{1}{K} \sum_{i=0}^{K-1} v_{\pi_i}(s)$.  This averaging of the value functions is used in our analysis to demonstrate that the returned mixture policy achieves the two feasibility objectives stated in sections 5 and 6.
>
> It's important to note that the mixture policy is a history-dependent policy. The notation $\frac{1}{K}\sum_{k=0}^{K-1} \pi_k$ is defined with respect to trajectories. Specifically, this means that the probability of a trajectory is the weighted average of the probabilities of that trajectory under each of the component policies. We will also clarify this in the paper.

---

> > ### Comment · Reviewer_ynfZ · 2024-08-08
> >
> > I thank the authors for their clarification on the differences between CAPI-QPI-PLAN and Confident-NPG-CMDP and the nature of $\bar{\pi}_K$. Can the authors kindly comment on questions 2, 3, 4.2 and 4.3 as well?

---

> ### Author Response · Authors · 2024-08-09
>
> Question 2: The complexity guarantees has worse dependence on the effective horizon e.g.
>  in Theorem 1 and in Theorem 2. Is this a consequence of the method or an artefact of the analysis? Can this be improved?
>
>  -----
>
> To address the strict-feasibility problem, the algorithm must solve a more conservative CMDP, as discussed in Section 6 of our paper. However, solving this conservative CMDP incurs a higher sample complexity cost, necessitating a separate treatment for the relaxed-feasibility setting. The worse dependence on the effective horizon is a consequence of the method.
>  Improving the effective horizon terms in both settings remains an area of ongoing research.
>
> -----
>
> Question 3: In Remark 3, the authors mention that the Slater's constant, which is required by their algorithm, can be approximated by another algorithm. Is this approximation trivial? How will the approximation error influence the current results?
>
> -----
>
> Recall that Slater's constant is defined as $\zeta \doteq \max\_{\pi} v^{c}\_{\pi}(s_0) - b$. To approximate $\zeta$, one can run CAPI-QPI-Plan against the local-access simulator with constraint function $c$, treating it as an unconstrained problem optimizing with respect to only $c$. This algorithm yields an approximation of $\max_{\pi} v^{c}_{\pi}(s_0)$, which can then be used to calculate $\zeta$. We can run CAPI-QPI-Plan to obtain an approximate $\zeta$ before executing Confident-NPG-CMDP, and the sample complexity of CAPI-QPI-Plan is only additive to that of Confident-NPG-CMDP.
>
> We will include the aforementioned elaboration on this remark in the paper.
>
> -----
>
> Question 4.2: What is a suitable choice of $c$ and how does tuning this parameter show up in the performance guarantees?
>
> -----
>
> We realized that the term $c$ was used in two different contexts, which may have caused some confusion. We used $c$ to denote both the constraint function and a user-defined parameter that bounds the importance sampling ratio in off-policy estimation, thereby determining the value of $m$. This overlap was an oversight on our part, and we will use two different notations to make the distinction in our paper. For the explanations that follow, we will refer to $c$ as the user-defined parameter that sets the value of $m$.
>
>
> By setting $c$ to a value greater than 0 adjusts the resampling window, controlled by the quantity $m = O(\ln(1+c) \cdot \text{poly}(\epsilon^{-1} (1-\gamma)^{-1}))$, to ensure that the off-policy value estimators are well-controlled. As a result, $c$ appears in $L = \lfloor K / (\lfloor m \rfloor + 1) \rfloor$, where $L$ is the total number of data sampling phases. Additionally, $c$ appears in $n$, the number of roll-outs for each state-action pair in a core set. In the proof of Theorem 1 in Appendix C (page 29), we show that the algorithm will make $nH(L+1)|\mathcal{C}_0|$ queries to the simulator.
>
>
> If $c > 0$, we see $L = \lfloor K / (\lfloor m \rfloor + 1) \rfloor \leq K/m \propto (1-\gamma)^{-3} \epsilon^{-1} (\ln(1+c))^{-1}$. All in all, we have $(1+c)^2/ \ln(1+c)$ showing up in the sample complexity.  Since $c$ is a constant, we have omitted it from the tilde-big-O notation in the final sample complexity.  In terms of $\epsilon$, with a factor of $\epsilon^{-2}$ contributed by $n$ and a factor of $\epsilon^{-1}$ contributed by $L$, we see that the total sample complexity is $\propto \epsilon^{-3}$.
>
>
> If $c$ is set to 0, then $\ln(1+c) = 0$ and $m = 0$, reducing $L$ to $\lfloor K \rfloor$. In this case, the algorithm reverts to a purely on-policy approach and results in a sample complexity $\propto \epsilon^{-4}$.  This motivated us to adopt the natural policy gradient and the off-policy approach to reduce the sample complexity from $\epsilon^{-4}$ to $\epsilon^{-3}$.
>
>
> A similar analysis for the strict-feasibility setting can be found in Theorem 2 in Appendix D (page 31).

---

> ### Author Response · Authors · 2024-08-09
> **Question 4.3: Can the authors comment on the memory requirement of Confident-NPG?**
>
> The overall memory requirement is $\tilde{d} n H L + \tilde{d} + L (m+1) \tilde{d} d$. The term $\tilde{d} n H L$ comes from maintaining $L+1$ copies of the core set $\mathcal{C}\_{l \in \\{0,...,L+1\\}}$, each containing no more than $\tilde{d}$ state-action pairs. For each state-action pair in $\mathcal{C}\_l$ for $l = 0,...,L$, the algorithm stores $n$ trajectories consisting of $H$ tuples $(s,a,r,c)$, requiring $\tilde{d} n H L$ memory. The additional $\tilde{d}$ accounts for the elements stored in $\mathcal{C}\_{L+1}$, which has no more than $\tilde{d}$ elements. In phase $L+1$, the algorithm terminates, so no roll-outs are required.
>
>
> The term $L (m+1) \tilde{d} d$ arises from storing the least-squares weights of the estimator when a core set is extended. When a core set is extended (i.e., when discovered = true), the least-squares weights for the corresponding $m+1$ iterations are recalculated for the extended set on line 20 and stored for that extension. With a maximum of $\tilde{d}$ extensions per core set and $m+1$ corresponding iterations for each core set, and $L$ core sets in total, each weight vector of dimension $d$, this results in the $L (m+1) \tilde{d} d$ memory bound for storing the least-squares weights.
>
>
> By substituting $n = \tilde{O}(\epsilon^{-2} (1-\gamma)^{-2})$, $H = \tilde{O}((1-\gamma)^{-1})$, $L = \tilde{O}((1-\gamma)^{-3} \epsilon^{-1})$, and $\tilde{d} = \tilde{O}(d)$, we obtain a total memory requirement of $\tilde{O}(d^2 \epsilon^{-3} (1-\gamma)^{-6})$.
>
>
> Below, we provide a formal discussion on why and how we store the least-squares weights.
>
>
> To return a mixture policy, the algorithm must access all policies $\pi_0,...,\pi_{K-1}$. Instead of storing the $\pi_k$ values for $k = 0,...,K-1$ across the entire state-action space, we store only the necessary information to reconstruct the policies when needed.
>
>
> For a phase $l$, the policies $\pi_k$ for $k \in \\{ k_l+1, \dots, k_{l+1} -1 \\}$ depend on the core set $\mathcal{C}_l$. Since $\mathcal{C}_l$ can be extended multiple times during the algorithm, we track newly added states in each extension and store the corresponding least-squares weights. $\mathcal{C}_0$ can only be extended via line 12, and any other $\mathcal{C}_l$ (for $l = 1,...,L+1$) only via line 27, when the running phase $\ell = l-1$. Newly added elements are marked at lines 12 and 27.  After lines 17-23 are executed, we store the least-squares weights associated with these newly added state-action pairs.
>
>
> Using the tuples stored in $\mathcal{C}\_l$, along with the marking of which state-action pairs are newly added in each extension and their associated least-squares weights, we can reconstruct the policy $\pi_{k+1}$. For example, let $\mathcal{C}_l^1 = \emptyset$, and $\mathcal{C}_l^i$ denote all state-action pairs added to $\mathcal{C}_l$ in extension $i$. Let $w^i_k$ represent the least-squares weights computed using $\mathcal{C}_l^1 \cup \mathcal{C}_l^2 \cup \dots \cup \mathcal{C}_l^i$ for the $k$-th iteration. When $\mathcal{C}_l$ is extended for the $(i+1)$-th time, let $\mathcal{C}_l^{i+1}$ be the newly added pairs, making the latest $\mathcal{C}\_l = \mathcal{C}\_l^1 \cup \mathcal{C}\_l^2 \cup \dots \cup \mathcal{C}\_l^{i+1}$. The least-squares weight $w^{i+1}\_k$ is then computed using $\mathcal{C}\_l$. When line 25 of algorithm 2 (pg 12) is executed, $\pi\_{k+1}$ remains unchanged for states already in $\text{Cov}(\mathcal{C}\_l^1 \cup \dots \cup \mathcal{C}\_l^i)$, equivalent to $\text{Cov}(\mathcal{C}\_{l+1})$ in line 25, because line 27 would have been executed in the previous extension $i$ making $\mathcal{C}\_{l+1} = \mathcal{C}\_l^1 \cup \dots \cup \mathcal{C}\_l^i$. For states in $\text{Cov}(\mathcal{C}\_l^1 \cup \dots \cup \mathcal{C}\_l^{i+1}) \setminus \text{Cov}(\mathcal{C}\_l^1 \cup \dots \cup \mathcal{C}\_l^i)$ (equivalent to $\text{Cov}(\mathcal{C}\_l) \setminus \text{Cov}(\mathcal{C}\_{l+1})$ in line 25), $\pi\_{k+1}$ makes an NPG update using $w^{i+1}\_k$. For all other states not in $\text{Cov}(\mathcal{C}\_l)$, the policy remains as $\pi_k$.
>
>
> A subroutine can perform these computations and return $\pi_{k}(\cdot | s)$ for any $s$ and $k$. By tracking newly added elements and the corresponding least-squares weights, the algorithm can reconstruct policies throughout its execution. From the stored data, the algorithm will have access to the constructed policies $\pi_0, \dots, \pi_{K-1}$ and return the value of a mixture policy when required.
>
>
> We plan to add this discussion to the appendix in the final version of our paper.

---

> > ### Comment · Reviewer_ynfZ · 2024-08-13
> >
> > Dear Authors,
> >
> > I appreciate the detailed responses to my questions, particularly those addressing the nature of $\pi_K$, the relevance of the off-policy evaluation scheme, the choice of $c$ (the user-defined parameter that sets the value of $m$), and the memory requirements of CAPI-NPG-CMDP. After carefully considering your response, I am convinced that CAPI-NPG-CMDP is a non-trivial extension of CAPI-QPI-PLAN,  which to my knowledge addresses the task of planning in Constrained Markov Decision Processes (CMDPs) under the weakest possible assumptions so far. On this note, I will revise my evaluation of your work.
> >
> > I strongly agree with including this discussion into the final version of the paper. I believe that doing so will enhance the readability of the paper and enable readers to better understand and appreciate the contribution of CAPI-NPG-CMDP in relation to CAPI-QPI-PLAN.
> >
> > Regarding future directions, while mixture policies have been generally accepted in the CMDP literature, I still believe they are not well-suited for the constrained MDP setting due to their nature of only ensuring good performance on average. On this note, I propose that the authors highlight this limitation in the discussion.
> >
> > Thank you again for your efforts in addressing my concerns.

---

> ### Author Response · Authors · 2024-08-13
>
> We are pleased to have addressed your concerns, and we sincerely thank the reviewer for their thoughtful questions.

---

### Author Rebuttal · Authors · 2024-08-05

We would like to begin by thanking all the reviewers for their time and dedication in evaluating our work. We start our rebuttal by emphasizing that our algorithm is not a straightforward extension of the work by Weisz et al. (2022).


First, we want to point out that the algorithm CAPI-QPI-Plan by Weisz et al. (2022) is designed for the unconstrained MDP setting, where it returns a deterministic policy. However, in the constrained MDP setting, an optimal deterministic policy for the unconstrained MDP may not be feasible. Thus, CAPI-QPI-Plan is not applicable to the constrained MDP setting.  In contrast, our algorithm, Confident-NPG-CMDP, returns a soft mixture policy $\bar \pi_{K}$, ensuring that $\bar \pi_K(a | s) > 0$ for all $(s, a) \in \mathcal{S} \times \mathcal{A}$. The soft policy returned by Confident-NPG-CMDP can solve the relaxed-feasibility problem in section 5 and the strict-feasibility problem in section 6.


The main algorithmic differences between Confident-NPG-CMDP and CAPI-QPI-Plan are as follows:
1) Data Sampling: Confident-NPG-CMDP does not sample data in every iteration, unlike CAPI-QPI-Plan.
2) Dual Variable Computation: Confident-NPG-CMDP requires computing the dual variable present in a primal-dual algorithm.
3) Policy Improvement Step: Confident-NPG-CMDP utilizes a softmax over the estimated action-values, whereas CAPI-QPI-Plan is greedy with respect to the estimated action-values.


These are critical changes to ensure a feasible mixture policy for the CMDP. Moreover, it also makes the analysis considerably more challenging.


We will now explain the motivation behind the changes to the policy improvement step and why we do not sample data in every iteration.  In the constrained MDP setting using a primal-dual approach, it is also crucial to control the dual variable. The dual variable is obtained using a mirror descent algorithm, introducing an additional $\epsilon^{-2}$ factor to the sample complexity. Simply applying CAPI-QPI-Plan and returning a mixture policy at the end would yield an overall sample complexity $\propto \epsilon^{-4}$, with the additional $\epsilon^{-2}$ factor arising from controlling the dual variable in addition to having to control for the estimation error. This complexity led us to adopt the natural policy gradient approach.


By using the natural policy gradient as a policy improvement step instead of CAPI-QPI-Plan's greedy policy improvement step, Confident-NPG-CMDP reduces the sample complexity from $\tilde O(\epsilon^{-4})$ to $\tilde O(\epsilon^{-3})$. This reduction is achieved by leveraging the softmax policy structure within the natural policy gradient, enabling effective use of off-policy estimation to conserve data. By employing a per-trajectory importance sampling ratio, we can weigh the Monte Carlo returns generated from data collected in an earlier on-policy phase, resulting in unbiased estimates of action values with respect to the target policy. However, this ratio can become large if there is a substantial difference between the on-policy and target policy. To address this, the algorithm collects data at intervals of $m$, effectively determining when to collect new data as the policy significantly diverges from the earlier on-policy data. By setting $c > 0$, we can bound the per-trajectory importance sampling ratio, thus controlling the interval $m = O(\ln(1+c) \text{poly}(\epsilon^{-1} (1-\gamma)^{-1}))$ for resampling on-policy data to produce well-controlled estimators.

---

### Decision · Program_Chairs · 2024-09-25

**Decision:**

Accept (poster)

**Comment:**

The paper proposes a new algorithm to solve constrained MDPs in the online infinite horizon setting under the q^pi-realizable and local access assumptions. There is general consensus among the reviewers of the interest of the results. Nonetheless, there was a concern about the technical novelty, since the algorithmic structure and the theoretical analysis heavily rely on existing results, in particular from the CMDP literature and on recent works such as CAPI-QPI-Plan. The authors have addressed these concerns in the rebuttal and explained how a direct application of CAPI-QPI-Plan would result in worse complexity and which ingredients of Confident-NPG-CMDP are key to achieve the desired results. Based on this, I believe the paper is significant and novel enough for the theoretical community working on CMDP as it pushes forward the limits of how we can solve this challenging problem. Hence, I recommend acceptance.

I strongly encourage the authors to improve the current manuscript as follows:
* Include a thorough discussion on the algorithmic and theoretical differences between CAPI-QPI-Plan and Confident-NPG-CMDP. This would greatly help in understanding the novel contributions of the paper.
* Highlight the limitations of the current work, notably the dependency on the horizon.
* Add a more comprehensive comparison and discussion with results in the linear MDP case and explain how moving to less restrictive assumptions makes the problem more challenging.
* Include the clarifications provided during the rebuttal.